# Elucidating the mitochondrial proteome of *Toxoplasma gondii* reveals the presence of a divergent cytochrome *c* oxidase

Azadeh Seidi[1†‡], Linden S Muellner-Wong[1†§], Esther Rajendran[1], Edwin T Tjhin[1], Laura F Dagley[2,3], Vincent YT Aw[1], Pierre Faou[4], Andrew I Webb[2,3], Christopher J Tonkin[2,3], Giel G van Dooren[1*]

[1]Research School of Biology, Australian National University, Canberra, Australia; [2]The Walter and Eliza Hall Institute of Medical Research, Victoria, Australia; [3]Department of Medical Biology, University of Melbourne, Victoria, Australia; [4]Department of Biochemistry and Genetics, La Trobe Institute for Molecular Science, La Trobe University, Victoria, Australia

**\*For correspondence:**
giel.vandooren@anu.edu.au

[†]These authors contributed equally to this work

**Present address:** [‡]Millennium Science, Victoria, Australia; [§]Department of Biochemistry and Molecular Biology, Monash Biomedicine Discovery Institute, Monash University, Melbourne, Australia

**Competing interests:** The authors declare that no competing interests exist.

**Abstract** The mitochondrion of apicomplexan parasites is critical for parasite survival, although the full complement of proteins that localize to this organelle has not been defined. Here we undertake two independent approaches to elucidate the mitochondrial proteome of the apicomplexan *Toxoplasma gondii*. We identify approximately 400 mitochondrial proteins, many of which lack homologs in the animals that these parasites infect, and most of which are important for parasite growth. We demonstrate that one such protein, termed *Tg*ApiCox25, is an important component of the parasite cytochrome *c* oxidase (COX) complex. We identify numerous other apicomplexan-specific components of COX, and conclude that apicomplexan COX, and apicomplexan mitochondria more generally, differ substantially in their protein composition from the hosts they infect. Our study highlights the diversity that exists in mitochondrial proteomes across the eukaryotic domain of life, and provides a foundation for defining unique aspects of mitochondrial biology in an important phylum of parasites.
DOI: https://doi.org/10.7554/eLife.38131.001

## Introduction

Mitochondria were derived from α-proteobacteria in an endosymbiotic event, and were present in the last common ancestor of all eukaryotes (*Gray, 2012*; *Sagan, 1967*). Mitochondria and related organelles are present in almost all extant eukaryotes. They function in the biosynthesis of molecules such as iron-sulfur clusters, lipids, coenzyme Q, heme and amino acids, the catabolism of molecules such as fatty acids, amino acids and monocarboxylates, the storage of ions and signalling molecules such as $Ca^{2+}$, the regulation of apoptosis, and the production of energy-carrying molecules such as ATP and NADH (*McBride et al., 2006*; *van Dooren et al., 2006*; *Zíková et al., 2016*). Notably, the function of mitochondria and related organelles vary substantially between different eukaryotic lineages, reflective of the >1500 M years that separate these lineages (*Chernikova et al., 2011*; *Zíková et al., 2016*).

Apicomplexans are a phylum of intracellular parasites that impose a significant medical and economic burden on human populations around the world. Apicomplexans include the causative agents of malaria (*Plasmodium* spp.), cryptosporidiosis (*Cryptosporidium* spp.) and toxoplasmosis (*Toxoplasma gondii*). *T. gondii* is an opportunistic parasite that chronically infects approximately one-third

of the world's adult population. Symptoms of *T. gondii* infection in healthy individuals are typically mild. In immunocompromised individuals and unborn fetuses, however, *T. gondii* infection can cause severe neurological and developmental impairment, and, without treatment, can lead to death of the infected individual (*Montoya and Liesenfeld, 2004*). The tractable genetics of *T. gondii* make it a versatile model for studying conserved aspects of apicomplexan biology.

Mitochondria are critical for the survival of apicomplexans, and several important drugs target proteins that function in this organelle, including atovaquone and endochin-like quinolones, both of which target cytochrome *c* reductase (complex III) of the electron transport chain, and DSM1, which targets dihydroorotate dehydrogenase, a central enzyme in pyrimidine biosynthesis (*Doggett et al., 2012*; *Phillips et al., 2008*; *Srivastava et al., 1999*). The functions of apicomplexan mitochondria are largely implied through comparative genomic approaches that have identified homologs of genes encoding known mitochondrial proteins from other eukaryotes (*Seeber et al., 2008*; *van Dooren et al., 2006*). The genome of apicomplexans such as *T. gondii* and *Plasmodium* spp. encode homologs of proteins involved in core mitochondrial processes such as the electron transport chain, the tricarboxylic acid (TCA) cycle, and the synthesis of molecules such as iron-sulfur clusters, heme and coenzyme Q (*Seeber et al., 2008*; *van Dooren et al., 2006*). The mitosome of *Cryptosporidium parvum* is highly reduced compared to that of other apicomplexans, lacking a TCA cycle, and harboring a minimal electron transport chain, but retaining the capacity for iron-sulfur cluster synthesis (*Mogi and Kita, 2009*).

Apicomplexans belong to a group of eukaryotes known as the myzozoans, that also include dinoflagellates and chromerids (*Cavalier-Smith and Chao, 2004*). Chromerids are a phylum of free-living, photosynthetic eukaryotes, and are thought to be the closest extant relatives of apicomplexans (*Moore et al., 2008*). All known myzozoans retain a mitochondrion or mitochondrion-derived organelle, the functions of which have numerous differences from well-studied eukaryotes such as yeast and animals (*Danne et al., 2013*; *Jacot et al., 2016*). In addition to a limited gene content, the mitochondrial genomes of these organisms contain unusually fragmented ribosomal RNAs (*Feagin et al., 2012*). Myzozoan mitochondria lack pyruvate dehydrogenase, the enzyme complex that decarboxylates pyruvate to form acetyl-CoA (*Danne et al., 2013*; *Foth et al., 2005*). Instead, these organisms have repurposed a branched-chain $\alpha$-ketoacid dehydrogenase (BCKDH) to catalyse this reaction (*Oppenheim et al., 2014*). Myzozoans have a functional TCA cycle, but some enzymes of this pathway are phylogenetically distinct from equivalent enzymes in yeast and humans (*Danne et al., 2013*; *Ke et al., 2015*; *MacRae et al., 2013*; *MacRae et al., 2012*; *van Dooren et al., 2006*). These organisms also contain a mitochondrial ATP synthase that lacks clear homologs to many of the proteins that comprise the membrane-bound $F_0$ component of the complex (*Balabaskaran Nina et al., 2011*; *Sturm et al., 2015*).

Elucidating the proteomes of mitochondria is key to understanding their functions. Organellar proteomics in intracellular parasites such as *T. gondii* is limited by available material and a lack of established organellar purification techniques. To overcome these obstacles, we adopted two complementary spatially-restricted biotin tagging approaches to define the proteome of the mitochondrial matrix of *T. gondii*. We identified over 400 putative mitochondrial matrix proteins, many of which have no ascribed function and no clear homology to proteins found in well-characterised eukaryotes such as yeast, and most of which are important for parasite growth and survival. We functionally characterise one protein that had no previously ascribed function, and demonstrate that this is a critical component of the cytochrome *c* oxidase (COX) complex of the mitochondrial electron transport chain in the parasite. We subsequently identify numerous apicomplexan-specific components of COX. These data reveal considerable divergence in the COX complex, and in mitochondria more generally, of *T. gondii* and related apicomplexans, compared to the animals they infect.

## Results

### Spatially-restricted biotinylation of mitochondrial matrix proteins

A genetically modified plant ascorbate peroxidase (APEX) was recently developed as a tool for biotinylating proximal proteins in mammalian cells (*Hung et al., 2014*; *Rhee et al., 2013*). This technique was used to define the proteome of the matrix and inter-membrane space of mammalian mitochondria. An alternative spatial biotinylation approach involves the use of a genetically modified,

promiscuous biotin protein ligase (BirA*), an approach that is typically utilised to elucidate protein-protein interactions (*Roux et al., 2012*). We sought to utilize APEX and BirA* to map the mitochondrial matrix proteome of *T. gondii*.

We generated parasite strains expressing APEX or BirA* fused at their N-termini to the mitochondrial matrix-targeting sequence of *Tg*Hsp60 (*van Dooren et al., 2009*). Immunofluorescence assays demonstrated that both mitochondrially-targeted APEX (mtAPEX) and mitochondrially-targeted

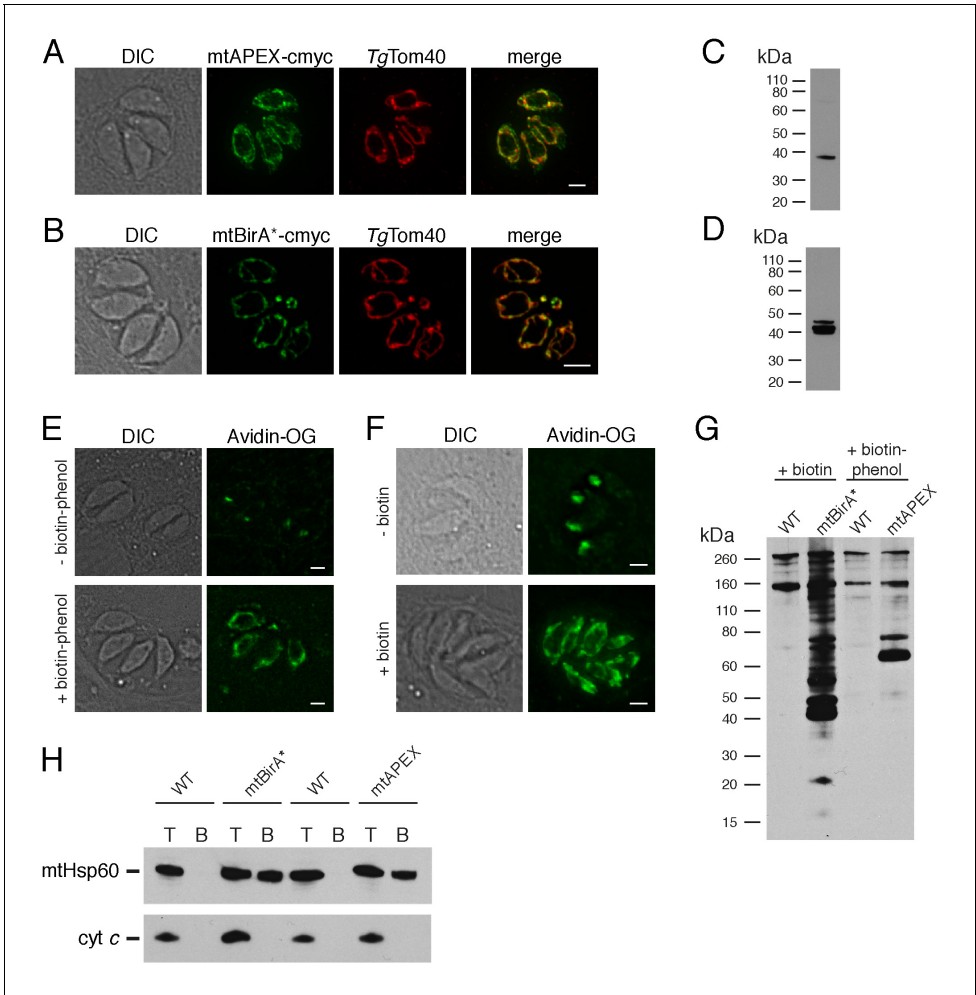

**Figure 1.** Biotinylation of mitochondrial matrix proteins in *T.gondii* parasites expressing mtAPEX and mtBirA*.T (**A–B**) Immunofluorescence assays of parasites expressing c-myc-tagged, mitochondrially-targeted APEX (**A**) and BirA* (**B**), labelled with anti-c-myc (green) and the mitochondrial marker *Tg*Tom40 (red). Scale bars are 2 μm. (**C–D**) Western blots of parasites expressing c-myc-tagged, mitochondrially-targeted APEX (**C**) and BirA* (**D**), labelled with anti-c-myc. (**E**) Oregon Green-conjugated avidin (Avidin-OG) labelling of *T. gondii* parasites expressing mtAPEX, and cultured in the absence (top) or presence (bottom) of biotin-phenol and $H_2O_2$. Biotinylated proteins are labelled in green. (**F**) Avidin-OG labelling of *T. gondii* parasites expressing mtBirA*, and cultured in the absence (top) or presence (bottom) of biotin. Biotinylated proteins are labelled in green. Scale bars are 2 μm. (**G**) Neutravidin-HRP protein blot of WT, mtBirA* or mtAPEX parasites cultured in the presence of biotin or biotin-phenol. (**H**) Western blots of the mitochondrial matrix marker mtHsp60 and the mitochondrial intermembrane space marker cyt *c* in WT, mtBirA* or mtAPEX parasites cultured in the presence of biotin (lanes 1 – 4) or biotin-phenol (lanes 5 – 8). Parasites were either harvested following treatment to yield the total (T) protein fraction, or biotinylated proteins were purified on a streptavidin-agarose column to yield the bound (B) fraction.

DOI: https://doi.org/10.7554/eLife.38131.002

The following figure supplement is available for figure 1:

**Figure supplement 1.** Map of the pBTM₃ plasmid vector, showing the *Avr*II, *Nde*I and *Not*I cut sites between which the APEX and BirA* cassettes were ligated (open reading frame of enzyme shown in green), the position of mitochondrial targeting leader sequence of *Tg*Hsp60 (Hsp60L; red), the 3x c-myc tag (yellow), and the positions of the phleomycin resistance marker (PhlR) for *T.gondii* selection, the ampicillin resistance marker for *E. coli* selection (AmpR), and the origin of replication (Ori; all magenta).

DOI: https://doi.org/10.7554/eLife.38131.003

BirA* (mtBirA*) co-localised with *Tg*Tom40, a marker for the *T. gondii* mitochondrion (*Figure 1A–B*; *van Dooren et al., 2016*). Western blots confirmed the presence of mtAPEX and mtBirA* proteins of the expected mass (*Figure 1C–D*).

To determine whether mtAPEX could label mitochondrial proteins, we treated parasites with biotin-phenol for 1 hr, initiated biotinylation by adding $H_2O_2$ for 1 min, then fixed and labelled parasites with Oregon green-conjugated avidin, a specific stain for biotinylated proteins. We observed mitochondrial labelling in treated parasites, and not in untreated parasites (*Figure 1E*), consistent with mtAPEX mediating the biotinylation of mitochondrial proteins. In untreated parasites, we observed labelling of endogenously biotinylated proteins in the apicoplast, consistent with previous observations (*Figure 1E*; *Chen et al., 2015*; *Jelenska et al., 2001*).

To determine whether mtBirA* could label mitochondrial proteins, we incubated mtBirA*-expressing parasites in medium supplemented with 1 mM biotin for 1 day. We labelled parasites with Oregon green-conjugated avidin and observed labelling in the mitochondrion of biotin-supplemented parasites, but not in untreated parasites (*Figure 1F*).

To observe the extent of protein biotinylation in the treated mtAPEX and mtBirA* parasites, we extracted proteins from RH strain wild type (WT), mtAPEX or mtBirA* parasites treated with either biotin-phenol and $H_2O_2$ or with biotin. We separated these by SDS-PAGE and probed with horse radish peroxidase (HRP)-conjugated neutravidin to label biotinylated proteins. In WT cells, we observed two major bands of the expected sizes of natively biotinylated proteins in these parasites (*Figure 1G*; *van Dooren et al., 2008*). In the biotin-phenol treated mtAPEX parasites, we observed labelling of several additional proteins, whereas in biotin-supplemented mtBirA* parasites, numerous proteins were labelled (*Figure 1G*). These data indicate that mtAPEX- and mtBirA*-mediated biotinylation is occurring in these parasites.

To determine the specificity of labelling, we extracted proteins from treated WT, mtAPEX and mtBirA* parasites and subjected these to affinity purification using streptavidin-conjugated magnetic beads. We separated purified proteins by SDS-PAGE and probed with antibodies against *Tg*Hsp60 (mtHsp60), a mitochondrial matrix marker (*Toursel et al., 2000*; *van Dooren et al., 2016*), and against *T. gondii* cytochrome *c* (cyt *c*), a mitochondrial intermembrane space marker (E.T. and G.v. D., unpublished). We did not detect mtHsp60 or cyt *c* in the streptavidin bound fraction in WT parasites treated with biotin-phenol and $H_2O_2$, or with biotin (*Figure 1H*). We detected bound mtHsp60, but not bound cyt *c*, in proteins extracted from both biotin-phenol-treated mtAPEX and biotin-treated mtBirA* parasites. This is consistent with the mitochondrial labelling that we observe being specific for the mitochondrial matrix.

## Quantitative proteomics to elucidate the mitochondrial matrix proteome

Having established two independent approaches for specifically labelling mitochondrial matrix proteins, we next undertook a label-free quantitative proteomic analysis of biotinylated proteins in treated mtAPEX and mtBirA* parasites. First, we generated three independent cell lysate pools of WT and mtAPEX cells treated with biotin-phenol and $H_2O_2$, and WT and mtBirA* cells treated with biotin. Biotinylated proteins were purified from these lysates using streptavidin beads, reduced, alkylated, and trypsin-digested before being identified using mass spectrometry (MS). Triplicate samples were then processed through our in-house quantitation pipeline to determine the relative abundance of each protein identified in the mtAPEX or mtBirA* samples as compared to WT controls. These data are represented on a volcano plot as a fold-change ($\log_2$ value) vs significance of the change ($-\log_{10}$ *p* value) (*Figure 2A–B*). This revealed an enrichment of numerous proteins in the mtAPEX and mtBirA* samples. Using cut-offs of p<0.001 and a WT:mtAPEX/mtBirA* $\log_2$ fold change of $\leq-2.5$, 421 proteins were identified in total: 213 proteins in the APEX samples and 369 proteins in the mtBirA* samples, with 161 proteins common to both proteomes (*Figure 2C*; *Supplementary file 1*). Hereafter, we refer to the list of 421 proteins as the mitochondrial proteome of *T. gondii*.

## Bioinformatic characterisation of the mitochondrial proteome

To test the validity of the *T. gondii* mitochondrial proteome, we undertook a series of in silico and experimental analyses. Proteins targeted to the mitochondrial matrix typically harbor an N-terminal

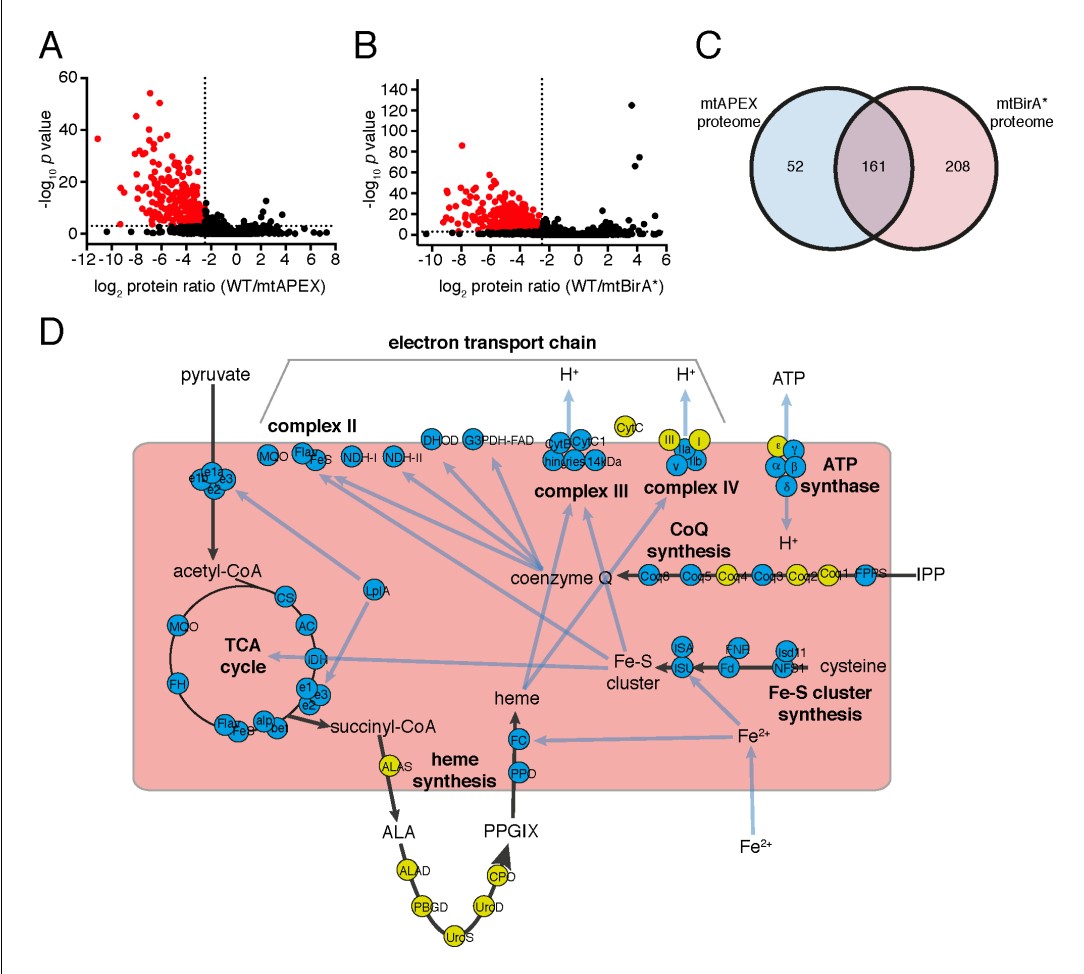

**Figure 2.** The mitochondrial proteome of *T.gondii*. (A–B) Volcano plots showing the $\log_2$ protein ratios vs $-\log10$ *p* values of biotinylated proteins in WT compared to mtAPEX (WT/APEX) samples (A) and in WT compared to mtbirA* samples (WT/BirA*) (B) following the quantitative pipeline analysis. Proteins were deemed to be enriched in the mitochondrion if the $\log_2$ fold change in protein expression was $\leq -2.5$ and the *p* value $\leq 0.001$ (red). (C) Venn diagram of the mtAPEX and mtBirA* proteomes. 161 proteins were identified in both proteomes, while 52 were unique to the mtAPEX proteome and 208 unique to the mtBirA* proteome. (D) Metabolic map of expected mitochondrial proteins (circles), showing proteins present (blue) and absent (yellow) from the *T. gondii* mitochondrial proteome. Black arrows represent the flow of metabolites through metabolic pathways in the mitochondrion, and blue arrows depict the flow of ions, minerals or metabolic pathway products.

DOI: https://doi.org/10.7554/eLife.38131.004

The following figure supplement is available for figure 2:

**Figure supplement 1.** Analysis of putative mitochondrial targeting peptides in the *T.gondii* mitochondrial proteome.

DOI: https://doi.org/10.7554/eLife.38131.005

amphipathic α-helix that facilitates import into the organelle (*van Dooren et al., 2016*). We examined the *T. gondii* mitochondrial proteome for proteins predicted to contain such an N-terminal targeting domain using the rules-based computational prediction tool MitoProt II (*Claros and Vincens, 1996*). Approximately 40% of proteins in the proteome had a strongly predicted N-terminal targeting sequences (probability of mitochondrial import >0.9), and a further ~20% had a moderately predicted targeting sequence (probability of import 0.5 – 0.9; *Figure 2—figure supplement 1*; *Supplementary file 1*). A further ~40% had a low probability of mitochondrial import (probability of import <0.5; *Figure 2—figure supplement 1*). This suggests that either the mitochondrial proteome has many false positives, or that many *T. gondii* mitochondrial proteins lack N-terminal targeting sequences, or that mitochondrial prediction tools such as MitoProt II are not suitable for analysis of mitochondrial proteins from *T. gondii*. Notably in this regard, the dataset used to develop MitoProt II did not include *T. gondii* proteins (*Claros and Vincens, 1996*).

Next, we examined metabolic pathway enrichment of the mitochondrial proteome (http://tox-odb.org; *Gajria et al., 2008*; *p* value cutoff < 0.05; *Supplementary file 2*). We observed the greatest enrichment in proteins involved in the TCA cycle, oxidative phosphorylation, and pyruvate metabolism, all processes predicted to occur in the mitochondrion. We observed enrichment of other processes predicted to occur in the mitochondrion, including ubiquinone biosynthesis, one carbon metabolism, and branched chain amino acid degradation. We also observed enrichment of some processes not expected to occur in the mitochondrion, such as glycolysis/gluconeogenesis, suggesting the presence of some false positives in our proteome.

To analyse the mitochondrial proteome in greater detail, we mapped proteins to previously constructed metabolic maps of apicomplexan mitochondria (*Supplementary file 3*; *Seeber et al., 2008*; *van Dooren et al., 2006*). This analysis identified all subunits of the BCKDH, all enzymes of the TCA cycle, all proteins predicted to function in mitochondrial Fe-S cluster synthesis, four of the seven enzymes predicted to function in coenzyme Q synthesis, and two of the three mitochondrial proteins involved in heme synthesis (*Figure 2D*). Additionally, we identified all five ubiquinone-reducing dehydrogenases of the mitochondrial inner membrane, and all currently predicted subunits of cytochrome *c* reductase (Complex III) and cytochrome *c* oxidase (Complex IV) that are encoded on the nuclear genome (*Figure 2D*). We were unable to identify cytochrome *b*, CoxI and CoxIII, proteins encoded on the mitochondrial genome of apicomplexan parasites. This is a surprising result. These proteins are all highly hydrophobic, and deeply embedded in the mitochondrial inner membrane as part of protein complexes. They may, therefore, not be accessible to biotinylation from the mitochondrial matrix. Alternatively, it is possible that due to the hydrophobic nature of these proteins, they were not available for enzymatic digestion, and thus failed to be detected by mass spectrometric analysis. A third possibility relates to the sequence of the mitochondrial genome in *T. gondii*, which has not yet been verified. Copies of the mitochondrial genome have migrated into the *T. gondii* nucleus, and (possibly pseudo) protein isoforms of these may have been included in the *T. gondii* protein dataset we used in identification. It is conceivable, therefore, that the predicted amino acid sequences of these do not match the sequence of the true proteins, which may have prevented their detection in our approaches. As expected, we did not identify the two isoforms of cytochrome *c*, both predicted to localize to the intermembrane space. We identified the α, β, γ and δ subunits of the $F_1$ component of ATP synthase, but not the ε subunit.

We identified numerous other proteins shown previously to localise to the mitochondrion (*Supplementary file 3*), including components of the presequence translocase associated motor (*van Dooren et al., 2016*), the aforementioned TgHsp60 (*Toursel et al., 2000*), components of the mitochondrial processing peptidase (*van Dooren et al., 2016*), a mitochondrial pyruvate kinase (*Saito et al., 2008*), a component of the γ−aminobutyric acid shunt (*MacRae et al., 2012*), an enzyme involved in phospholipid synthesis (*Hartmann et al., 2014*), and proteins associated with DNA repair (*Garrison and Arrizabalaga, 2009*) and antioxidant defences (*Brydges and Carruthers, 2003*; *Ding et al., 2004*). Notably, we were unable to identify mitochondrially-localised enzymes involved in the 2-methylcitrate cycle (*Limenitakis et al., 2013*). We identified several conserved solute transporter proteins, and numerous proteins with housekeeping roles in the mitochondrion, including mitochondrial RNA polymerase, 18 ribosomal proteins (*Gupta et al., 2014*), ribosome maturation factors, and various translation elongation factors (*Supplementary file 3*). A recent genome-wide CRISPR-based screen to identify genes important for in vitro growth of tachyzoite-stage *T. gondii* identified 15 so-called Indispensable Conserved Apicomplexan Proteins (ICAPs; *Sidik et al., 2016*). Of these, eight localised to the mitochondrion. We identified 6 of these mitochondrial ICAPs in the mitochondrial proteome. In total, our proteomics identified 97 out of 111 proteins that previous studies had experimentally localized, or predicted to localize, to the mitochondrion, suggesting a high level of coverage (87%).

We analysed which of the 97 'true positive' mitochondrial proteins were found in both proteomes, and which were identified in the mtAPEX or mtBirA* proteomes alone (*Supplementary file 3*). We found that 54% of the 97 true positive proteins were identified in both proteomes, and that 19% and 28% were identified in only the mtAPEX or mtBirA* proteomes, respectively. We compared these percentages to the overall percentages of proteins found in each dataset (*Figure 2C*). Notably, the 28% of 'expected' mitochondrial proteins identified in the mtBirA*-alone category is an under-representation of the 49% (208/421) of proteins in this category overall (*Figure 2C*). This

raises the possibility that proteins categorised in the mtBirA* proteome alone contain more false positives than in the shared and mtAPEX-alone categories.

We examined the mitochondrial proteome for likely false positives. We identified 16 proteins that other studies have demonstrated do not localise to the mitochondrion (*Supplementary file 3*). Notably, 14 of these were identified only in the mtBirA* proteome, while one was present in both the mtBirA* and mtAPEX proteomes, again consistent with the existence of more false positives in the mtBirA* proteome. In total, of the 421 proteins in the mitochondrial proteome, we identified 97 that are known or expected to localise to the mitochondrion, and 16 that are known to localise elsewhere.

## Localisation of uncharacterized proteins from the mitochondrial proteome

Of the 421 putative mitochondrial proteins identified in the mitochondrial proteome, 150 (36%) were annotated as 'hypothetical' proteins, and a further 140 (33%) had no previously defined role or experimentally determined localization in *T. gondii* (*Supplementary file 1*). We attempted to localize 37 proteins selected at random from this 'uncharacterized' protein data set by introducing a hemagglutinin (HA) epitope tag at the 3' end of the native locus of genes encoding these proteins. We then undertook immunofluorescence assays to determine the localization of the proteins, co-labelling with anti-*Tg*Tom40 as a marker for the mitochondrion (*van Dooren et al., 2016*). We were successful in localizing 27 of the 37 selected proteins. Of these, 22 (81%) localized to the mitochondrion, three to the cytosol, one to the endoplasmic reticulum and one to the nucleus (*Figure 3*). All 13 that were identified in both the mtAPEX and mtBirA* proteomes localized to the mitochondrion, suggesting a high degree of confidence in the mitochondrial localization of proteins identified from both datasets. Five of the six proteins found solely in the mtAPEX proteome localized to the mitochondrion, while four of the eight proteins found only in the mtBirA* proteome localized to the mitochondrion. This is consistent with the dataset of proteins identified only in the mtBirA* proteome having more false positives than the other datasets.

## Phylogenetic analyses of the mitochondrial proteome

We next examined the evolutionary history of proteins from the mitochondrial proteome. First, we undertook reciprocal Basic Local Alignment Search Tool (BLAST) searches to identify homologs of proteins from the mitochondrial proteome in the apicomplexan parasites *Plasmodium falciparum*, *Babesia bovis* and *Cryptosporidium parvum*, and the chromerid *Vitrella brassicaformis*. Using this approach, we identified homologs for 71% of *T. gondii* mitochondrial proteins in *P. falciparum*, 61% in *B. bovis*, 28% in *C. parvum,* and 83% in *V. brassicaformis* (*Figure 4A*).

We were next interested in the extent of novelty in the *T. gondii* mitochondrial proteome when compared to non-apicomplexan eukaryotes. We examined conserved orthology groupings of the 421 proteins in the mitochondrial proteome and identified 418 proteins that clustered into 412 separate orthology groups (http://orthomcl.org; *Chen et al., 2006*). We identified 86 proteins that were unique to *T. gondii* and closely related coccidians such as *Neospora caninum*, 243 proteins with orthologs in non-apicomplexan eukaryotes, and a set of 89 proteins that were found only in apicomplexans and/or chromerids (*Figure 4B*; *Supplementary file 1*).

Novel drug targets against apicomplexans are likely to emerge from proteins which lack homologs in animals. We therefore conducted an orthology analysis comparing the *T. gondii* mitochondrial proteome to other apicomplexans, chromerids and animals. We found that 51% of the mitochondrial proteome lacked orthologs in animals, of which 56% had orthologs in other apicomplexans and/or chromerids (*Figure 4C*). The remainder were restricted to *T. gondii* and other coccidian parasites such as *N. caninum*.

In an additional analysis, we searched for homologs of proteins from the *T. gondii* mitochondrial proteome in a recently-published 'high-confidence' mitochondrial proteome of yeast (*Morgenstern et al., 2017*). We identified yeast homologs to 161 proteins in the *T. gondii* mitochondrial proteome (*Supplementary file 1*). Of these, 103 were predicted to localize to the yeast mitochondrion, and 58 were predicted not to localize to the yeast mitochondrion. The high proportion of non-mitochondrial homologs suggests that aspects of mitochondrial biology in *T. gondii* may localize elsewhere in other eukaryotes. Notably, however, 42 of the 58 'non-mitochondrial' proteins

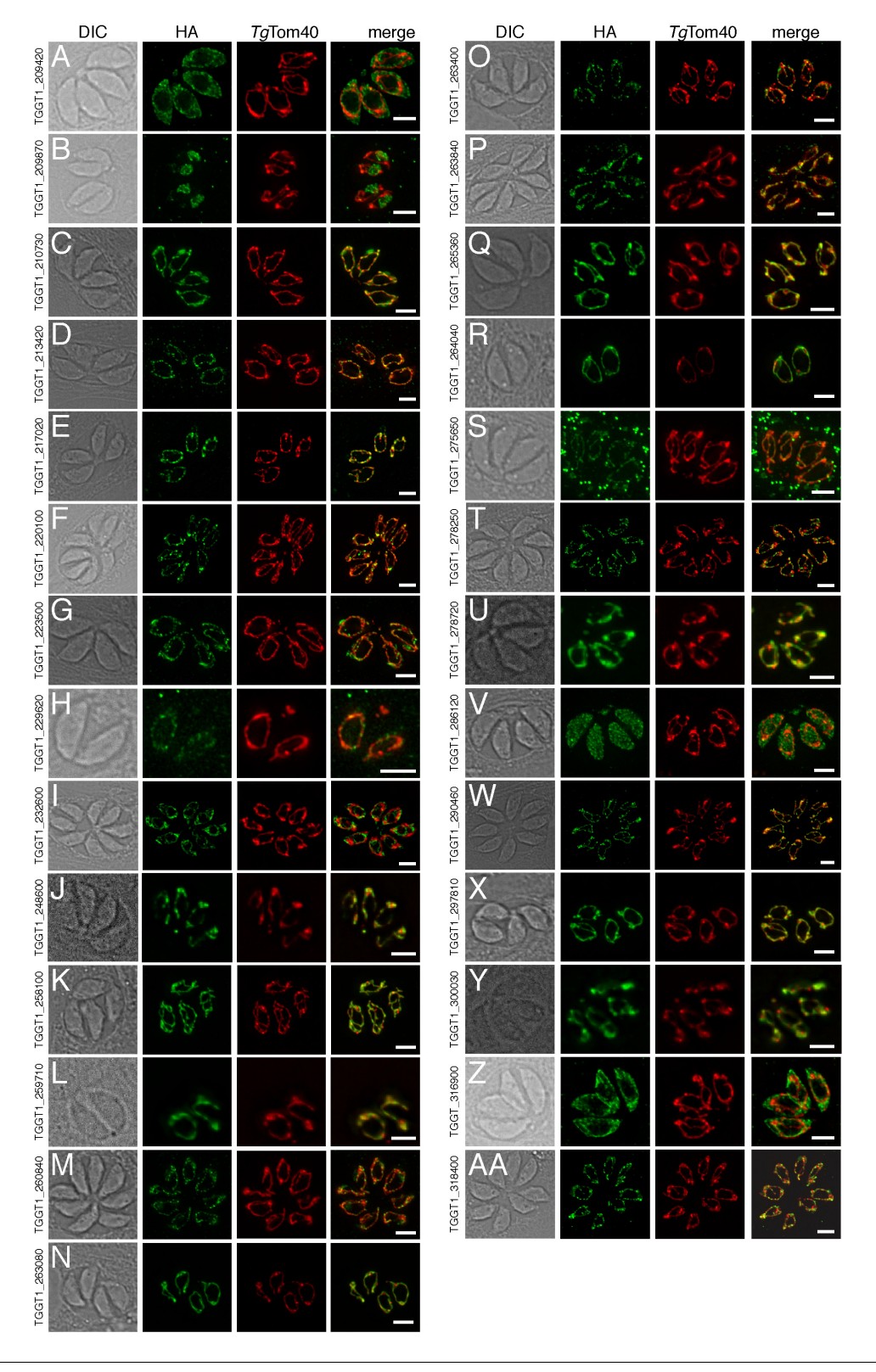

**Figure 3.** The localization of novel proteins from the *T.gondii* mitochondrial proteome. (**A–AA**) Proteins with no previously determined localization in *T. gondii* were selected from the mitochondrial proteome, and the corresponding gene was tagged at the 3'-terminus of the open reading frame with a HA tag. Immunofluorescence assays depict HA-tagged proteins (green) co-labelled with the mitochondrial marker *Tg*Tom40 (red). The http://toxodb. org gene identification number is depicted for every gene that was tagged.
*Figure 3 continued on next page*

*Figure 3 continued*

DOI: https://doi.org/10.7554/eLife.38131.006

The following figure supplement is available for figure 3:

**Figure supplement 1.** Map of the pgCH plasmid vector, showing the *Spe*I, *Bgl*II and *Avr*II cut sites between which the 3' flanks of target genes were ligated (green), the position of the 1x HA tag (yellow), and the positions of the chloramphenicol resistance marker (Chl$^R$) for *T.gondii* selection, the ampicillin resistance marker for *E. coli* selection (Amp$^R$), and the origin of replication (Ori; all magenta).

DOI: https://doi.org/10.7554/eLife.38131.007

(72%) were identified in the BirA*-only dataset, which our previous analyses suggest may harbor more false positives. It is conceivable, therefore, that many of the proteins with non-mitochondrial homologs in yeast are false positives.

## Phenotype analyses of the mitochondrial proteome

Our data indicate substantial novelty in mitochondrial biology of *T. gondii* and related organisms. Often, such derived features are less important for an organism's survival than proteins that have been conserved across evolution. A recent genome-wide, CRISPR-based loss-of-function screen in *T. gondii* found that genes conserved in eukaryotes were, in general, more important for parasite

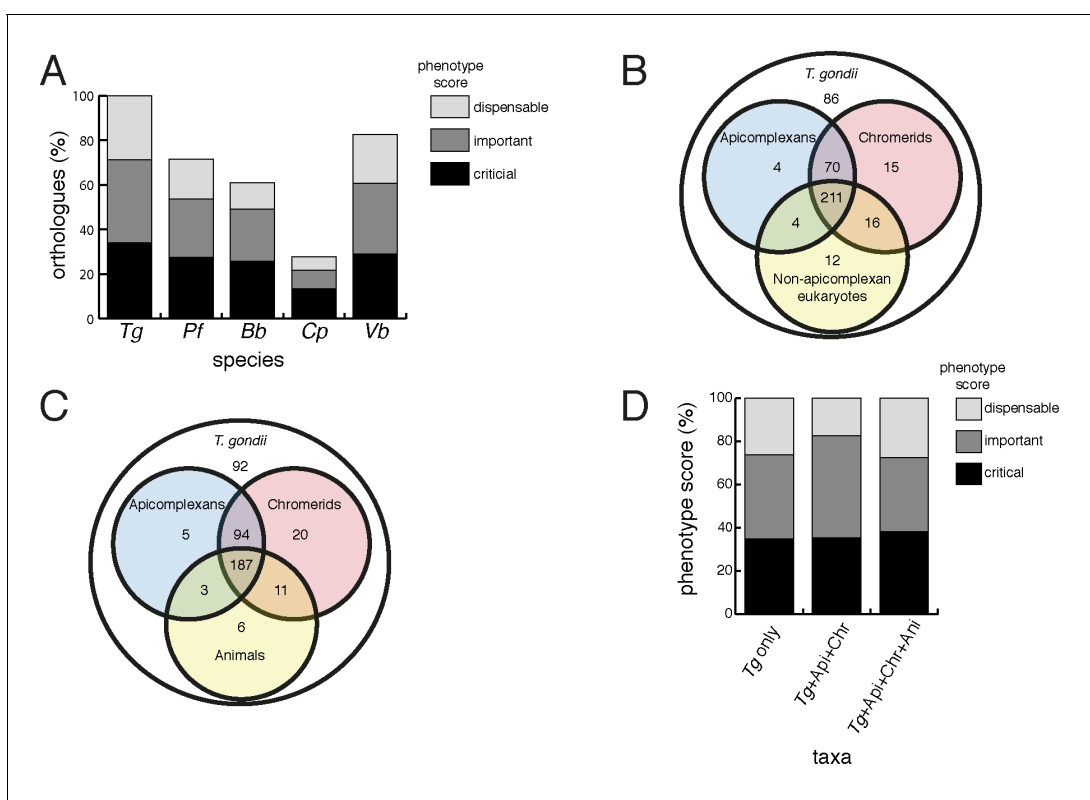

**Figure 4.** Orthology analyses of proteins from the *T.gondii* mitochondrial proteome reveal that many mitochondrial proteins are restricted to *T. gondii* and related organisms, and that most are important for parasite survival. (**A**) Bar graph depicting the percentage of orthologs from the mitochondrial proteome of *T. gondii* (*Tg*) found in *P. falciparum* (*Pf*), *B. bovis* (*Bb*), *C. parvum* (*Cp*) and *V. brassicaformis* (*Vb*). Phenotype scores are indicated with shading, and reveal that most ortholog groups in each category are important or critical for tachyzoite growth. (**B–C**) Venn diagram depicting ortholog groupings from the mitochondrial proteome of *T. gondii* compared to (**B**) non-coccidian apicomplexans, chromerids and eukaryotes, or (**C**) non-coccidian apicomplexans, chromerids and animals. (**D**) Bar graph depicting distribution of phenotype scores in genes belonging to ortholog groups found only in *T. gondii* and other coccidians (*Tg* only), in *T. gondii*, non-coccidian apicomplexans and chromerids (*Tg*+ Api+ Chr), and in *T. gondii*, non-coccidian apicomplexans, chromerids and animals (*Tg*+ Api+ Chr+ Api). In (**A**) and (**D**), genes with phenotype scores of >-2 were considered dispensable, −2 to −4 were considered important, and <-4 were considered critical.

DOI: https://doi.org/10.7554/eLife.38131.008

fitness than genes with a more restricted phylogenetic distribution (*Sidik et al., 2016*). To determine whether the same was true for mitochondrial proteins, we analysed the *T. gondii* mitochondrial proteome using the Sidik et al data set (*Sidik et al., 2016*). The Sidik et al study ascribed phenotype scores to each gene in the *T. gondii* nuclear genome, with more negative scores indicative of a greater importance for a gene's contribution to parasite fitness. The Sidik et al study found that most genes that were important for parasite growth had phenotype scores of below −2, and most dispensable genes had phenotype scores of greater than −2 (*Sidik et al., 2016*). Based on this, we categorised proteins in the mitochondrial proteome as dispensable (phenotype score >-2), important (−2 to −4), or critical (<-4) for parasite growth. Notably, 35% of proteins from the mitochondrial proteome were critical, and 39% were important, for parasite growth (*Figure 4A*; *Supplementary file 1*). Of the *T. gondii* mitochondrial proteins with orthologs in *P. falciparum*,~75% were important or critical, and similar values apply for those with orthologs in *B. bovis*, *C. parvum*, and *V. brassicaformis* (*Figure 4A*). Of the proteins conserved between *T. gondii* and apicomplexans/chromerids, over 80% were important or critical for parasite growth, while over 70% of proteins collectively found in apicomplexans, chromerids and animals were important or critical (*Figure 4D*; *Supplementary file 1*).

## *Tg*ApiCox25 is important for mitochondrial oxygen consumption in *T. gondii*.

Having identified ~175 proteins in the *T. gondii* mitochondrial proteome that have no clear orthologs outside the apicomplexan/chromerid lineage, and no predicted function, we embarked on a broad project to characterise the importance and role of these proteins. In the remainder of this manuscript, we focus on one such protein, annotated as TGGT1_264040, which (for reasons that will become apparent) we termed *Tg*ApiCox25. *Tg*ApiCox25 belongs to an OrthoMCL ortholog grouping that is restricted to apicomplexans, contains no recognisable functional domains, and is important for parasite fitness. It has a predicted molecular mass of 25 kDa, and we confirmed its localisation to the mitochondrion (*Figure 3R*). To establish the importance of *Tg*ApiCox25 for parasite growth, and to facilitate subsequent characterisation of its function, we replaced the native promoter of *Tg*ApiCox25 with an anhydrotetracycline (ATc)-regulated promoter using a CRISPR-based genome editing approach (*Figure 5—figure supplement 1A*). We performed PCR screening analysis to identify clonal parasites that had integrated the ATc-regulated promoter into the *Tg*ApiCox25 locus (*Figure 5—figure supplement 1B–C*). We termed the resultant ATc-regulated *Tg*ApiCox25 strain 'r*Tg*ApiCox25'. We then introduced a HA tag at the 3' end of the open reading frame of the r*Tg*ApiCox25 locus. We termed the resultant HA-tagged, ATc-regulated *Tg*ApiCox25 strain 'r*Tg*ApiCox25-HA'. To measure the extent of target protein knockdown upon the addition of ATc in the r*Tg*ApiCox25 strain, we cultured parasites in the absence of ATc, or in the presence of ATc for 1 – 3 days, then undertook western blotting. This revealed substantial depletion of *Tg*ApiCox25-HA 2 days after ATc addition, with the *Tg*ApiCox25-HA protein barely detectable after 3 days in ATc (*Figure 5A*). To determine the importance of *Tg*ApiCox25 on parasite growth we compared plaque sizes of parental wild type (WT) and r*Tg*ApiCox25 parasites grown in the absence or presence of ATc for 9 days. This revealed that growth of r*Tg*ApiCox25, but not WT, parasites was severely impaired in the presence of ATc (*Figure 5B–C*). Interestingly, r*Tg*ApiCox25 parasites grew better in the absence of ATc than WT parasites (*Figure 5B–C*). To determine whether the growth phenotype observed upon *Tg*ApiCox25 knockdown was specifically due to loss of *Tg*ApiCox25, we complemented the r*Tg*ApiCox25 strain with a copy of *Tg*ApiCox25 expressed from the constitutive α-tubulin promoter (generating a strain we termed c*Tg*ApiCox25-HA/r*Tg*ApiCox25). The presence of the constitutive copy of *Tg*ApiCox25 restored growth of r*Tg*ApiCox25 parasites in the presence of ATc (*Figure 5B–C*).

A major function of the mitochondrion is in oxidative phosphorylation, where the catabolism of organic molecules by the TCA cycle and other metabolic pathways contributes electrons to an electron transport chain on the inner membrane of the organelle (*van Dooren et al., 2006*). Electrons are ultimately used to reduce $O_2$, with the electron transport chain simultaneously generating a proton gradient across the inner membrane. This proton gradient is then used to drive the F-type ATP synthase, a rotary motor that phosphorylates ADP to form ATP, the energy currency of cells. Defects in any of the processes involved in oxidative phosphorylation will lead to defects in mitochondrial $O_2$ consumption. To test whether *Tg*ApiCox25 has a role in oxidative phosphorylation, we established

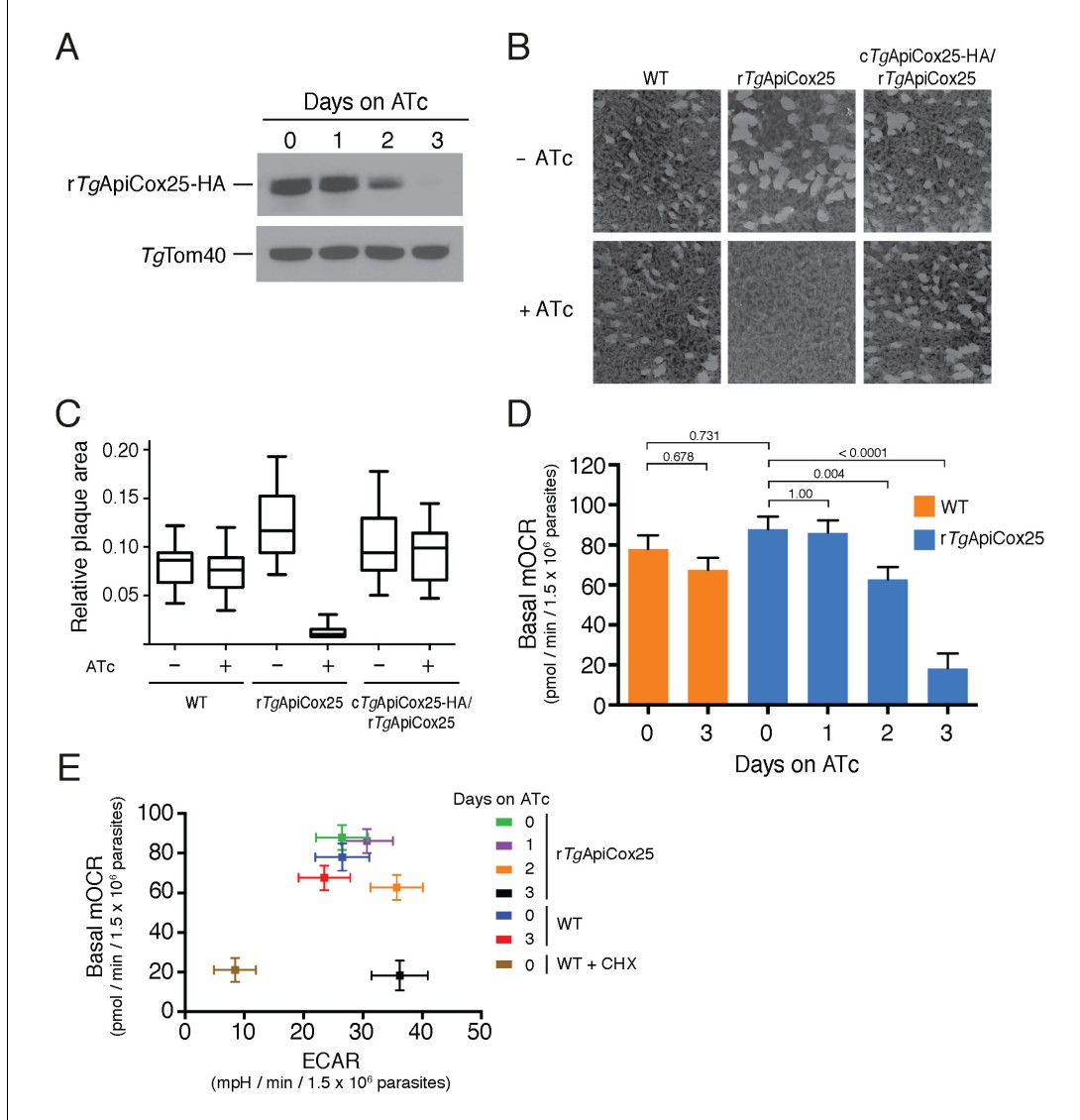

**Figure 5.** *Tg*ApiCox25 is important for parasite growth and mitochondrial O$_2$ consumption. (**A**) Western blot of proteins extracted from r*Tg*ApiCox25-HA parasites grown in the absence of ATc, or in ATc for 1–3 days, and detected using anti-HA antibodies (top) and anti-*Tg*Tom40 (as a loading control; bottom). (**B**) Plaque assays measuring growth of WT, r*Tg*ApiCox25 and complemented c*Tg*ApiCox25-HA/r*Tg*ApiCox25 parasites cultured in the absence (top) or presence (bottom) of ATc. Assays are from a single experiment and are representative of 3 independent experiments. (**C**) Quantification of plaque size from WT, r*Tg*ApiCox25 and complemented c*Tg*ApiCox25-HA/r*Tg*ApiCox25 parasites grown in the absence or presence of ATc for 9 days. Box and whisker plots depict the median plaque size (centre line), the 25[th] and 75[th] percentiles (box) and the 5[th] and 95[th] percentiles (lines). Data are from 30 plaques per flask from a single experiment, except in the case of the r*Tg*ApiCox25 strain, where only 18 plaques were discernible. (**D**) Basal mitochondrial oxygen consumption rates (mOCR) in WT parasites grown in the absence of ATc or in the presence of ATc for 3 days (orange), and r*Tg*ApiCox25 parasites grown in the absence of ATc, or in the presence of ATc for 1 – 3 days (blue). A linear mixed-effects model was fitted to the data, and the values depict the mean ± s.e.m. from three independent experiments. A one-way ANOVA followed by Tukey's multiple pairwise comparison test was performed. Relevant *p* values are shown. (**E**) Basal mOCR plotted against basal extracellular acidification rate (ECAR) of WT cells grown in the absence of ATc, the presence of cycloheximide (CHX) for 1 day, or the presence of ATc for 3 days, and r*Tg*ApiCox25 parasites grown in the absence of ATc or presence of ATc for 1 – 3 days (mean ± s.e.m. of the linear mixed-effects model described above; n = 3).
DOI: https://doi.org/10.7554/eLife.38131.009

The following figure supplements are available for figure 5:

**Figure supplement 1.** Generating an ATc-regulated promoter replacement strain of *Tg*ApiCox25.
DOI: https://doi.org/10.7554/eLife.38131.010

**Figure supplement 2.** Knockdown of *Tg*ApiCox25 leads to defects in maximal mOCR.
DOI: https://doi.org/10.7554/eLife.38131.011

*Figure 5 continued on next page*

*Figure 5 continued*

**Figure supplement 3.** Defects in mOCR upon *Tg*ApiCox25 knockdown are not the result of general defects in mitochondrial morphology or parasite viability.

DOI: https://doi.org/10.7554/eLife.38131.012

**Figure supplement 4.** Map of the pUgCTH₃ plasmid vector, showing the *Bgl*II and *Avr*II cut sites between which the *Tg*ApiCox25 open reading frame was ligated (green), the position of the 3x HA tag (yellow) and α-tubulin 5′ region (blue), and the positions of the chloramphenicol resistance marker (Chl^R) for *T.gondii* selection, the UPRT flank (linearized at the indicated *Mfe*I site before transfection), the ampicillin resistance marker for *E. coli* selection (Amp^R), and the origin of replication (Ori; all magenta).

DOI: https://doi.org/10.7554/eLife.38131.013

an assay to measure $O_2$ consumption by the parasite using a Seahorse XFe96 extracellular flux analyzer. We grew r*Tg*ApiCox25 parasites in the absence of ATc, or presence of ATc for 1 – 3 days then used the XFe96 analyzer to measure basal mitochondrial $O_2$ consumption rates (mOCR) in extracellular parasites. This revealed a significant,~80% depletion in basal mOCR upon *Tg*ApiCox25 knockdown (**Figure 5D**), concomitant with knockdown of protein levels (**Figure 5A**).

Treatment with the protonophore FCCP uncouples OCR from ATP synthesis and enables the determination of maximal mOCR in parasites. We found that maximal mOCR was also depleted upon *Tg*ApiCox25 knockdown (**Figure 5—figure supplement 2**). In a separate study, we demonstrated that depletion of a component of *T. gondii* ATP synthase led to an increase in the spare capacity of mOCR (i.e. the difference between the basal and maximal mOCR; **Huet et al., 2018**). We did not observe an increase in spare capacity upon *Tg*ApiCox25 knockdown (**Figure 5—figure supplement 2**), indicating that *Tg*ApiCox25 is not a component of ATP synthase. Basal and maximal mOCR in WT parasites were unaffected by the addition of ATc, although, curiously, the spare capacity in WT parasites was greater than in r*Tg*ApiCox25 parasites grown in the absence of ATc (**Figure 5—figure supplement 2**). This is perhaps reflective of changes in *Tg*ApiCox25 protein abundance or timing of expression upon replacing the native promoter with the ATc-regulated promoter.

We wondered whether the defect in mOCR upon *Tg*ApiCox25 knockdown was the result of general defects in mitochondrial function or parasite viability. To address this, we first measured the extracellular acidification rate (ECAR) in WT and r*Tg*ApiCox25 parasites grown in the absence or presence of ATc. ECAR is measured simultaneously with OCR by the XFe96 analyzer. In mammalian cells, ECAR is thought to depend on the extrusion of lactate, and is therefore a measure of glycolysis (**Ferrick et al., 2008**). The contribution of glycolysis and other processes that acidify the extracellular medium (e.g. activity of the plasma membrane proton pump of these parasites; **Moreno et al., 1998**) to ECAR in *T. gondii* are not yet understood. Nevertheless, we can use ECAR measurements as a general indication of parasite metabolic activity. ECAR levels in WT and r*Tg*ApiCox25 parasites grown in the absence of ATc was approximately 27 mpH/min/1.5 × 10⁶ parasites. Growth of *Tg*ApiCox25 parasites for 2 or 3 days in ATc resulted in a slight increase in ECAR (**Figure 5E**), indicating that parasites remained metabolically active upon *Tg*ApiCox25 knockdown. As a control for non-metabolically active parasites, we treated WT parasites with the translation inhibitor cycloheximide for 24 hr, which would be expected to deplete key metabolic enzymes in the parasite. XFe96 measurements revealed that both mOCR and ECAR were depleted upon cycloheximide treatment (**Figure 5E**), consistent with a general loss of parasite metabolism leading to simultaneous defects in mOCR and ECAR. We conclude that parasites remain metabolically active in the absence of *Tg*ApiCox25.

Next, we asked whether knockdown of *Tg*ApiCox25 led to general defects in mitochondrial morphology. We performed immunofluorescence assays labelling the mitochondrion in r*Tg*ApiCox25 parasites grown in the absence or presence of ATc. This revealed no gross morphological defects in mitochondria upon the loss of *Tg*ApiCox25 (**Figure 5—figure supplement 3A**). Finally, we asked whether *T. gondii* parasites remained viable upon knockdown of *Tg*ApiCox25. We pre-incubated r*Tg*ApiCox25 parasites in the presence of ATc for 3 days. We then set up plaque assays in the absence or presence of ATc, comparing parasite growth with parasites that had not been pre-incubated in ATc. As expected, parasites that were maintained in ATc for the duration of the plaque assay underwent minimal growth, regardless of whether they were pre-incubated in ATc (**Figure 5—figure supplement 3B**). Notably, plaque number and size were equivalent between pre-incubated

and non-pre-incubated parasites when grown in the absence of ATc. This reveals that *Tg*ApiCox25 knockdown is reversible and, importantly, that r*Tg*ApiCox25 parasites treated for 3 days on ATc have equivalent viability to r*Tg*ApiCox25 not grown on ATc.

The data presented here indicate that the defects we observed in mitochondrial $O_2$ consumption upon *Tg*ApiCox25 knockdown were not due to general defects in parasite viability, metabolism or mitochondrial morphology. We conclude that *Tg*ApiCox25 has an important, specific role in oxidative phosphorylation in *T. gondii* parasites.

## *Tg*ApiCox25 is a component of cytochrome *c* oxidase

Our findings that *Tg*ApiCox25 is critical for mitochondrial $O_2$ consumption prompted us to investigate whether this protein is a component of the mitochondrial electron transport chain that mediates $O_2$ consumption. The mitochondrial electron transport chain consists of several large protein complexes. To determine whether *Tg*ApiCox25 exists in a protein complex, we extracted proteins from *Tg*ApiCox25-HA parasites using 1% (v/v) Triton X-100 detergent, and separated these proteins by blue native-PAGE, a technique that preserves the native conformation of proteins and protein complexes. Western blotting of *Tg*ApiCox25-HA extracts separated by blue native-PAGE and detected with anti-HA antibodies revealed that *Tg*ApiCox25-HA exists at a molecular mass of ~600 kDa (*Figure 6A*). By contrast, the monomeric form of *Tg*ApiCox25-HA, extracted from parasites and separated by SDS-PAGE, had a mass of approximately 25 kDa (*Figure 6B*). We conclude that *Tg*ApiCox25 is a component of a ~ 600 kDa protein complex in the parasite mitochondrion.

To elucidate the proteins that comprise the *Tg*ApiCox25-containing complex, we immunoprecipitated *Tg*ApiCox25-HA and associated proteins with anti-HA-coupled agarose beads (*Figure 6—figure supplement 1A*), then performed mass spectrometry to identify the proteins that were part of this complex. As a negative control, we immunopurified *Tg*Tom40-HA (*Figure 6—figure supplement 1B*), the central protein of the ~400 kDa translocon of the outer mitochondrial membrane (TOM) complex (*van Dooren et al., 2016*), and subjected these extracts to mass spectrometry-based protein identification. Using this approach, we identify 12 proteins, including *Tg*ApiCox25, that were enriched in the *Tg*ApiCox25-HA immunoprecipitation compared to the *Tg*Tom40-HA immunoprecipitation (*Figure 6C*; *Table 1*; *Supplementary file 5*). Of these 12 proteins, three are annotated as being canonical components of cytochrome *c* oxidase (COX, also known as Complex IV of the mitochondrial electron transport chain; *Figure 6C*; *Table 1*). These proteins included *Tg*Cox2A (TGGT_226590) and *Tg*Cox2b (TGGT1_310470), split Cox2 homologs that are found in apicomplexans and which have previously been localised to the mitochondrion of *T. gondii* (*Funes et al., 2002*; *Morales-Sainz et al., 2008*), as well as *Tg*Cox5b (TGGT1_209260). Profile hidden Markov model similarity searches revealed that 7 of the remaining 8 *Tg*ApiCox25-associated proteins have homologs in other apicomplexans as well as in chromerids, whereas one (TGGT1_265370; *Tg*ApiCox16) is restricted to *T. gondii* (*Table 1*). All of the proteins detected were identified in the mitochondrial proteome (*Table 1*; *Supplementary file 1*), including TGGT1_297810 (*Tg*ApiCox30), an HA-tagged version of which we previously confirmed localizes to the mitochondrion in immunofluorescence assays (*Figure 3X*). All these proteins except *Tg*ApiCox16 are predicted to be important for growth of the tachyzoite stage of *T. gondii* (*Table 1*; *Sidik et al., 2016*).

Three further proteins, TGGT1_254030, TGGT1_242840, and a cytochrome *c* oxidase subunit III (CoxIII) homologue (TGVEG_442760), were highly enriched in the *Tg*ApiCox25 immunoprecipitation but excluded from our analysis because they were absent from at least one replicate of the control data set (*Supplementary file 5*; *Table 1*). Both TGGT1_254030 and TGGT1_242840 were present in the mitochondrial proteome, and were predicted to be important for parasite growth. TGGT1_242840 was phylogenetically restricted to apicomplexans and chromerids, whereas TGGT1_254030 had homology to the human CDGSH iron-sulfur domain containing protein 3 (*Table 1*).

As a direct test for whether *Tg*ApiCox25 interacts with *Tg*Cox2a, we introduced a FLAG epitope tag into the native locus of *Tg*Cox2a in the *Tg*ApiCox25-HA background strain, generating a strain we termed *Tg*Cox2a-FLAG/*Tg*ApiCox25-HA (*Figure 7—figure supplement 1A–B*). We separated proteins from the *Tg*Cox2a-FLAG/*Tg*ApiCox25-HA strain using blue native-PAGE and performed western blotting with anti-FLAG antibodies. This revealed that *Tg*Cox2a-FLAG exists in a protein complex of ~600 kDa (*Figure 7A*). Immunoprecipitation of *Tg*Cox25-HA with anti-HA antibodies co-purified *Tg*Cox2a-FLAG, but not *Tg*AtpB, the β–subunit of the $F_1$ domain of ATP synthase, or the

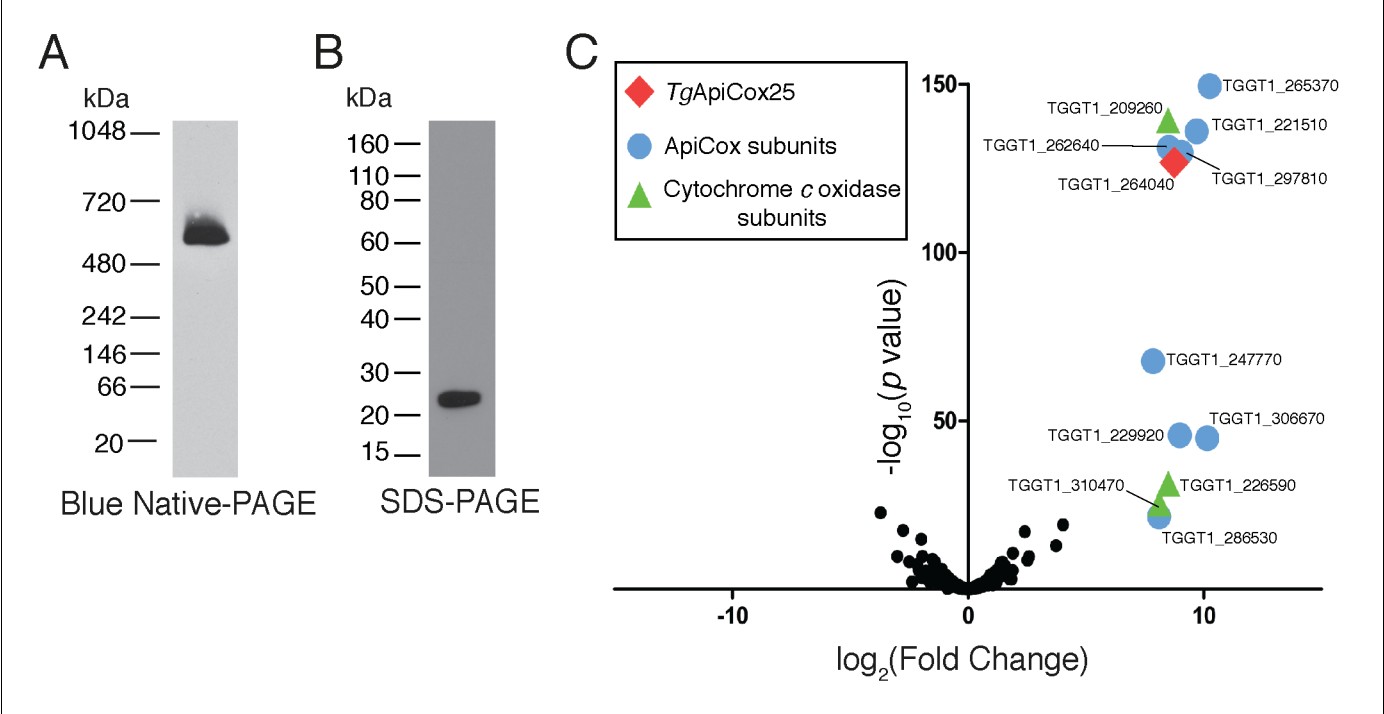

**Figure 6.** *Tg*ApiCox25 is part of a 600 kDa protein complex and co-purifies with canonical components of the cytochrome *c* oxidase complex. (**A**) Western blot of proteins extracted from *Tg*ApiCox25-HA parasites, separated by blue native-PAGE, and detected with anti-HA antibodies. (**B**) Western blot of proteins extracted from *Tg*ApiCox25-HA parasites, separated by SDS-PAGE, and detected with anti-HA antibodies. (**C**) Volcano plot showing the $\log_2$ fold change vs $-\log_{10} p$ values of proteins purified from *Tg*ApiCox25-HA vs *Tg*Tom40-HA parasites using anti-HA immunoprecipitations and detected by mass spectrometry. Only proteins detected in each of the three independent experiments for both parasite lines are depicted. Proteins enriched in the *Tg*ApiCox25-HA samples ($p<0.05$; $\log_2$ fold change >5) have been coded according to whether they are orthologous to canonical cytochrome *c* oxidase subunits (green triangles), or restricted to the apicomplexan lineage (blue circles; ApiCox subunits). *Tg*ApiCox25 is also depicted (red diamond).

DOI: https://doi.org/10.7554/eLife.38131.014

The following figure supplement is available for figure 6:

**Figure supplement 1.** Immunopurification of the *Tg*ApiCox25 and *Tg*Tom40 protein complexes.
DOI: https://doi.org/10.7554/eLife.38131.015

mitochondrial outer membrane protein *Tg*Tom40 (*Figure 7B*). Similarly, immunoprecipitation of *Tg*Cox2a-FLAG with anti-FLAG antibodies co-purified *Tg*ApiCox25, but not *Tg*AtpB or *Tg*Tom40 (*Figure 7B*). Together, these data indicate that *Tg*Cox2a-FLAG exists in the same complex as *Tg*ApiCox25-HA.

To test whether *Tg*ApiCox25 interacts with *Tg*ApiCox30, we introduced a FLAG epitope tag into the native locus of TgApiCox25 in the *Tg*ApiCox30-HA background, generating a strain we termed *Tg*ApiCox25-FLAG/*Tg*ApiCox30-HA (*Figure 7—figure supplement 1C–D*). Western blots of proteins separated by blue native-PAGE indicated that both *Tg*ApiCox25-FLAG and *Tg*ApiCox30-HA exist in a ~600 kDa protein complex (*Figure 7C*). Immunoprecipitation of *Tg*ApiCox30 with anti-HA antibodies purified *Tg*ApiCox25 but not *Tg*AtpB or *Tg*Tom40, and immunoprecipitation of *Tg*ApiCox25 with anti-FLAG antibodies co-purified *Tg*ApiCox30 but not *Tg*AtpB or *Tg*Tom40 (*Figure 7D*). We conclude that *Tg*ApiCox25-FLAG and *Tg*ApiCox30-HA exist in the same protein complex.

Combined with the observation that *Tg*ApiCox25 is important for mitochondrial $O_2$ consumption (*Figure 5D*), these data are consistent with the hypothesis that *Tg*ApiCox25 and *Tg*ApiCox30 are components of the parasite COX complex, the terminal oxidase of the mitochondrial electron transport chain that facilitates the reduction of $O_2$. To reflect their apparent phylogenetic restriction to apicomplexans and related organisms, we have termed the apicomplexan-specific components of the *T. gondii* COX complex as 'ApiCox' proteins, with the numerical suffix indicating the predicted molecular mass of the protein (*Table 1*).

**Table 1.** Summary of the features of proteins identified in proteomic analysis of the *TgApiCox25* complex.
Similarity searches were performed using HMMER (https://www.ebi.ac.uk/Tools/hmmer/). The accession numbers listed were derived from http://EuPathDB.org (apicomplexan and chromerid species) or www.ncbi.nlm.nih.gov (all others). Abbreviations: *Plasmodium falciparum (Pf)*, *Cryptosporidium parvum (Cp)*, *Vitrella brassicaformis (Vb)*, *Saccharomyces cerevisiae (Sc)*, *Homo sapiens (Hs)*, *Arabidopsis thaliana (At)*.

| ToxoDB gene ID (http://toxodb.org) | Protein annotation | Predicted protein mass (kDa) | Mitochondrial proteome (this study) | Phenotype score (Sidik et al., 2016) | Pf | Cp | Vb (Chromerid) | Sc (Fungi) | Hs (Animal) | At (Plant) |
|---|---|---|---|---|---|---|---|---|---|---|
| TGGT1_264040 | Hypothetical protein (*TgApiCox25*) | 24.5 | ✓ | −2.54 | Conserved unknown protein PF3D7_1464000.1 (2.4e$^{-53}$) | x | Hypothetical Protein Vbra_12326 .t1 (5.2e$^{-29}$) | x | x | x |
| TGGT1_265370 | Hypothetical protein (*TgApiCox16*) | 16.0 | ✓ | 1.56 | x | x | x | x | x | x |
| TGGT1_209260 | Putative cytochrome *c* oxidase subunit (*TgCox5b*) | 34.8 | ✓ | −3.07 | Putative COX5B PF3D7_0927800.1 (4.3e$^{-101}$) | x | COX5B-2 Vbra_9355 .t1 (1.6e$^{-92}$) | Cox4p P04037 (0.32) | Cox5B NP_001853.2 (0.05) | COX5b At1g80230 (2.1e$^{-08}$) |
| TGGT1_221510 | Hypothetical protein (*TgApiCox18*) | 17.9 | ✓ | −3.28 | Conserved unknown protein PF3D7_0523300.1 (1.5e$^{-48}$) | x | Hypothetical Protein Vbra_21271 .t1 (5.2e$^{-45}$) | x | x | x |
| TGGT1_262640 | Cg8 family protein (*TgApiCox23*) | 23.8 | ✓ | −3.49 | Cg8 protein PF3D7_0708700.1 (3.1e$^{-64}$) | x | Hypothetical Protein Vbra_3012 .t1 (2.4e$^{-53}$) | x | x | x |
| TGGT1_297810 | Hypothetical protein (*TgApiCox30*) | 29.6 | ✓ | −3.64 | Conserved unknown protein PF3D7_0915700.1 (1.2e$^{-46}$) | x | Hypothetical Protein Vbra_17445 .t1 (6.7e$^{-33}$) | x | x | x |
| TGGT1_247770 | Hypothetical protein (*TgApiCox19*) | 19.2 | ✓ | −2.61 | Conserved unknown protein PF3D7_1402200.1 (1.2e$^{-34}$) | x | Hypothetical Protein Vbra_2065 .t1 (1.7e$^{-27}$) | x | x | x |
| TGGT1_229920 | Hypothetical protein (*TgApiCox35*) | 35.0 | ✓ | −3.84 | Conserved unknown protein PF3D7_0306500.1 (1.5e$^{-90}$) | x | Hypothetical Protein Vbra_6819 .t1 (1.6e$^{-73}$) | x | x | x |
| TGGT1_306670 | Hypothetical protein (*TgApiCox26*) | 25.8 | ✓ | −3.68 | Conserved unknown protein PF3D7_1439600.1 (2.6e$^{-43}$) | x | Hypothetical Protein Vbra_888 .t1 (1.2e$^{-36}$) | x | x | x |
| TGGT1_226590 | Putative cytochrome *c* oxidase subunit (*TgCox2a*) | 34.5 | ✓ | −3.80 | Cytochrome c oxidase subunit 2 PF3D7_1361700.1 (4.9e$^{-58}$) | x | Cytochrome c oxidase subunit 2 Vbra_8641 .t1 (3.6e$^{-33}$) | Cox2 P00410 (2.6e$^{-06}$) | Cox2 P00403 (0.0004) | Cox2 P93285 (3.3e$^{-06}$) |
| TGGT1_310470 | Putative cytochrome *c* oxidase subunit (*TgCox2b*) | 21.2 | ✓ | −4.18 | Cytochrome c oxidase subunit 2 PF3D7_1430900.1 (7.6e$^{-75}$) | x | Cytochrome c oxidase subunit 2 Vbra_14923 .t1 (4.2e$^{-7}$) | Cox2 P00410 (4.3e$^{-31}$) | Cox2 P00403 (9.2e$^{-29}$) | Cox2 P93285 (3.8e$^{-37}$) |
| TGGT1_286530 | Hypothetical protein (*TgApiCox24*) | 25.4 | ✓ | −2.82 | Conserved unknown protein PF3D7_1362000.1 (6.0e$^{-45}$) | x | Hypothetical Protein Vbra_10089 .t1 (1.2e$^{-11}$) | x | x | x |

*Table 1 continued on next page*

*Table 1 continued*

| ToxoDB gene ID (http://toxodb.org) | Protein annotation | Predicted protein mass (kDa) | Mitochondrial proteome (this study) | Phenotype score (Sidik et al., 2016) | Similarity search (E-value) | | | | | |
|---|---|---|---|---|---|---|---|---|---|---|
| | | | | | Pf | Cp | Vb (Chromerid) | Sc (Fungi) | Hs (Animal) | At (Plant) |
| TGGT1_254030 | Zinc finger CDGSH-type domain-containing protein (*Tg*ApiCox13) | 13.2 | ✓ | −4.26 | CDGSH iron-sulfur domain-containing protein PF3D7_1022900.1 ($7.8e^{-42}$) | x | CDGSH iron-sulfur domain-containing protein 3 Vbra_4701 .t1 ($1.2e^{-44}$) | x | CDGSH iron-sulfur domain-containing protein 3 ($2.7e^{-11}$) | x |
| TGGT1_242840 | Membrane protein (*Tg*ApiCox14) | 13.9 | ✓ | −3.58 | Conserved unknown protein PF3D7_1339400.1 ($4.1e^{-16}$) | x | Hypothetical Protein Vbra_9996 .t1 ($1.8e^{-9}$) | x | x | x |
| TGVEG_442760 | Cytochrome C family oxidase subunit III (*Tg*CoxIII) | 16.8 | - | N/A | Cytochrome c oxidase subunit 3 mal_mito_1 ($5.2e^{-5}$) | x | x | x | x | x |

DOI: https://doi.org/10.7554/eLife.38131.016

To investigate whether proteins from the *Tg*ApiCox25 complex might have structural similarity to known proteins, we queried each protein enriched in the *Tg*ApiCox25 immunoprecipitation against the Protein Data Bank (PDB) using HHPRED, a profile hidden Markov model search tool that also incorporates secondary structure information (*Zimmermann et al., 2018*). As expected, *Tg*Cox2a, *Tg*Cox2b, *Tg*CoxIII, and *Tg*Cox5b had homology to equivalent cytochrome *c* oxidase proteins from other eukaryotes (probability >98.9%, e value <$6.6 e^{-11}$ for each). Interestingly, *Tg*ApiCox25 had predicted homology to cytochrome *c* oxidase complex subunit 6A from *Bos taurus* (PDB annotation 5B1A_T; probability 90.9%, e value 0.58) and *Tg*ApiCox23 had predicted homology to cytochrome *c* oxidase complex subunit four from *Bos taurus* (PDB annotation 5B1A_D; probability 95.57%, e value 0.0057). Of the remaining ApiCox proteins, ApiCox13 was predicted to have homology to the mitochondrial matrix-localized CDGSH iron-sulfur domain containing protein 3 of humans (PDB annotation 6AVJ_C; probability 99.95, e value $3.1 e^{-31}$), an iron-sulfur cluster-containing NEET-family protein important for mitochondrial iron homeostasis (*Lipper et al., 2018*). None of the remaining ApiCox proteins were matched to proteins from PDB with any great confidence (probability of homology to the top 'hit'<70%, e value >5 for all).

To obtain insights into the role of *Tg*ApiCox25 in the parasite COX complex, we tested the effects of *Tg*ApiCox25 knockdown on complex integrity. We introduced a FLAG epitope tag into the *Tg*Cox2a locus of the r*Tg*ApiCox25-HA parasite strain, generating a strain we term *Tg*Cox2a-FLAG/r*Tg*ApiCox25-HA (*Figure 7—figure supplement 1A–B*). We grew parasites in the absence of ATc or the presence of ATc for 1 – 3 days, then separated protein extracts by SDS-PAGE. As demonstrated previously, ATc treatment led to depletion of *Tg*ApiCox25 but not of a *Tg*Tom40 loading control (*Figure 7E*). Interestingly, knockdown of *Tg*ApiCox25 also led to depletion of *Tg*Cox2a-FLAG, although not to the same extent as *Tg*ApiCox25 (*Figure 7E*). We solubilised proteins from the *Tg*ApiCox25-HA/*Tg*Cox2a-FLAG strain in 1% (v/v) Triton X-100 and separated proteins by blue native PAGE. Knockdown of *Tg*ApiCox25 led to depletion of the ~600 kDa COX complex (*Figure 7F*). Interestingly, *Tg*ApiCox25 knockdown resulted in the appearance of a ~ 400 kDa complex that contains *Tg*Cox2a (*Figure 7D*). Together, these observations are consistent with *Tg*ApiCox25 having an important role COX complex integrity, stability and/or assembly.

## Discussion

In this study, we utilised two spatially-restricted biotin tagging approaches to elucidate the proteome of the mitochondrial matrix of *T. gondii*. These complementary approaches identified approximately 400 putative mitochondrial proteins. This number is slightly lower than the 495 proteins identified in the mtAPEX-derived proteome of mammalian cells (*Rhee et al., 2013*), and is less than

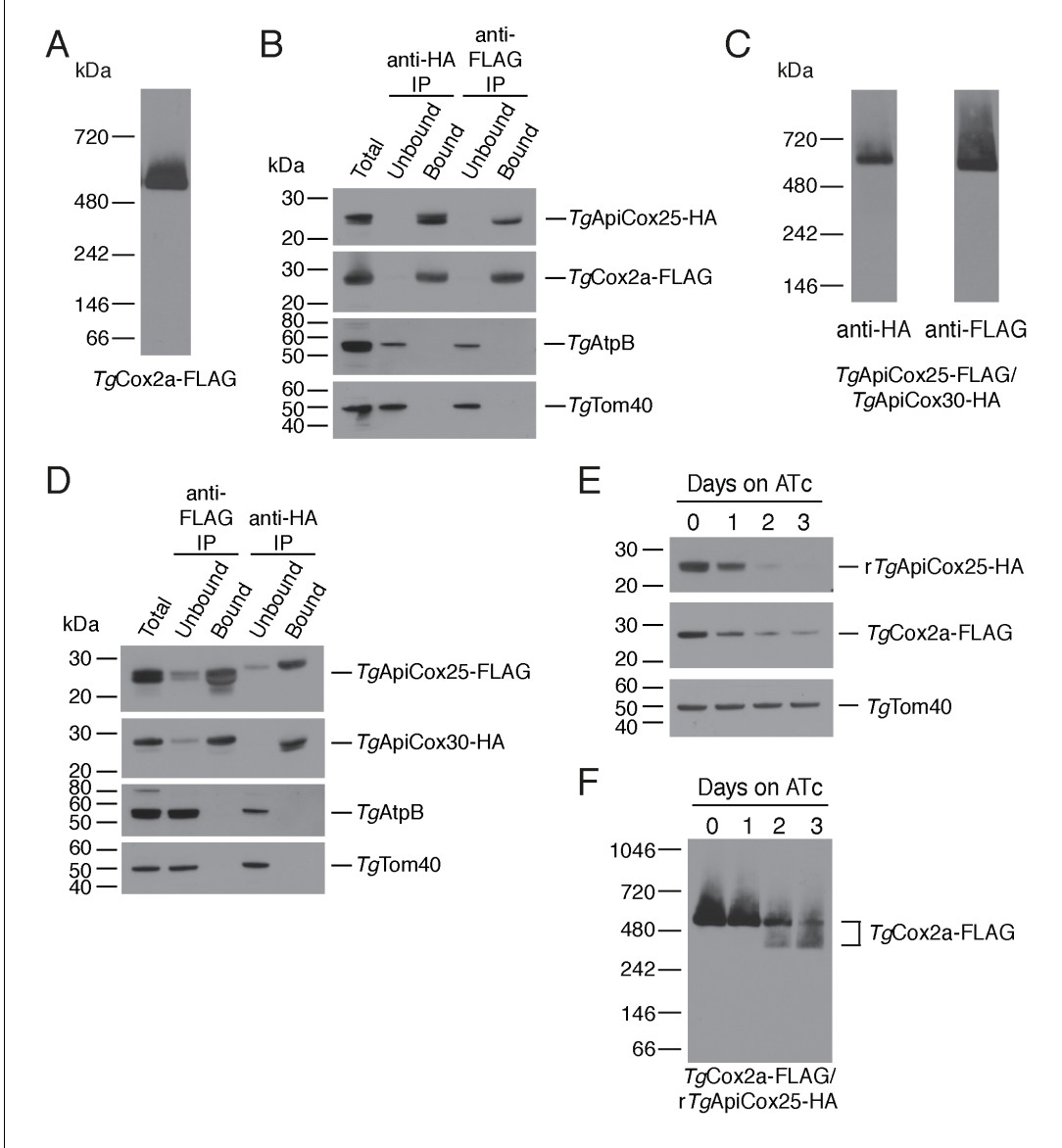

**Figure 7.** *Tg*ApiCox25 is a component of *T.gondii* cytochrome *c* oxidase and important for complex integrity. (**A**) Anti-FLAG western blot of proteins from the *Tg*Cox2a-FLAG/*Tg*ApiCox25-HA strain separated by blue native-PAGE. (**B**) Western blots of proteins extracted from the *Tg*Cox2a-FLAG/ *Tg*ApiCox25 HA strain and subjected to immunoprecipitation using anti-HA (anti-HA IP) or anti-FLAG (anti-FLAG IP) antibody-coupled beads. Extracts include samples before immunoprecipitation (Total), samples that did not bind to the anti-HA or anti-FLAG beads (Unbound), and samples that bound to the anti-HA or anti-FLAG beads (Bound). Samples were probed with anti-HA to detect *Tg*ApiCox25-HA, anti-FLAG to detect *Tg*Cox2a-FLAG, anti-AtpB to detect the β-subunit of *T. gondii* ATP synthase, and anti-*Tg*Tom40. (**C**) Anti-HA (left) and anti-FLAG (right) western blots of proteins from the *Tg*ApiCox25-FLAG/*Tg*ApiCox30-HA strain separated by blue native-PAGE. (**D**) Western blots of proteins extracted from the *Tg*ApiCox25-FLAG/ *Tg*ApiCox30-HA strain and subjected to immunoprecipitation using anti-HA (anti-HA IP) or anti-FLAG (anti-FLAG IP) antibody-coupled beads. Extracts include samples before immunoprecipitation (Total), samples that did not bind to the anti-HA or anti-FLAG beads (Unbound), and samples that bound to the anti-HA or anti-FLAG beads (Bound). Samples were probed with anti-HA to detect *Tg*ApiCox30-HA, anti-FLAG to detect *Tg*ApiCox25-FLAG, anti-AtpB, and anti-*Tg*Tom40. (**E**) Western blot of proteins extracted from r*Tg*ApiCox25-HA/*Tg*Cox2a-FLAG parasites grown in the absence of ATc, or in ATc for 1 – 3 days, separated by SDS-PAGE and detected using anti-HA (top), anti-FLAG (middle) and anti-*Tg*Tom40 (as a loading control; bottom). (**F**) Western blot of proteins extracted from *Tg*Cox2a-FLAG/r*Tg*ApiCox25-HA parasites grown in the absence of ATc, or in ATc for 1 – 3 days, separated by blue native-PAGE, and detected using anti-FLAG antibodies.

DOI: https://doi.org/10.7554/eLife.38131.017

The following figure supplement is available for figure 7:

**Figure supplement 1.** Generating FLAG tagged *Tg*Cox2a and *Tg*ApiCox25 strains.

DOI: https://doi.org/10.7554/eLife.38131.018

the 750 – 900 proteins identified in yeast studies using highly purified mitochondria (*Morgenstern et al., 2017*; *Sickmann et al., 2003*). Our proteome identified most of the proteins 'expected' to localize to the *T. gondii* mitochondrion (*Figure 2D*; *Supplementary file 3*), suggesting a high level of coverage. The lowered numbers of proteins in the *T. gondii* mitochondrial proteome compared to the yeast mitochondrial proteome may represent reduced functions in the parasite organelle compared to that in a metabolically flexible organism such as yeast, although our subsequent analyses (discussed below) are consistent with the presence of a high degree of divergent biology in apicomplexan mitochondria.

We experimentally localized 27 previously uncharacterised proteins from the *T. gondii* mitochondrial proteome, finding that 22 of these localised to the mitochondrion (*Figure 3*). From this, we estimate that ~80% of the 'uncharacterized' proteins from the proteome localize to the mitochondrion. Our findings suggest a low false positive rate in the proteins identified from the APEX proteome alone (~5%), but a higher false positive rate in the mtBirA* proteome (~20%). Based on these analyses, we consider the 213 proteins from the mtAPEX and shared mtAPEX/mtBirA* proteomes to be 'likely' mitochondrial proteins, and the 208 proteins found in the mtBirA* proteome alone to be 'possible' mitochondrial proteins.

To our knowledge, our study is the first time the APEX and BirA* restricted biotinylation approaches have been directly compared in organellar proteomic approaches. Bioinformatic and localization analyses indicate a high level of overlap between the two approaches, but also indicate the presence of unique proteins in both data sets. This suggests that the two approaches provide complementary information that expands the organellar proteome, while at the same time providing confidence in the shared set of proteins that were identified. The mtBirA*-derived proteome identified greater numbers of proteins, while also featuring greater numbers of false positives.

The mtAPEX labelling was performed over a short timeframe in extracellular parasites, whereas the mtBirA* labelling was performed across 24 hr in intracellular parasites. The differences we see in these proteomes may, in part, result from differences in the labelling conditions. In this regard, it is notable that of the 20 'expected' mitochondrial ribosomal protein and ribosome assembly factors, 17 were only present in the (intracellular) mtBirA* proteome (*Supplementary file 3*). Future work could examine whether there are differences in mitochondrial ribosome assembly and abundance between intra- and extra-cellular parasites. Similarly, mitochondrial carrier proteins, which mediate the transport of solutes into and out of the mitochondria, were enriched in the APEX proteome (5 of the six carrier proteins identified were present in the 'APEX only' proteome, and the other in both proteomes; *Supplementary file 1*). This may reflect either differences in the abundance of these proteins in intra- vs extra-cellular parasites, or in different labelling affinities for these integral membrane proteins between biotin-phenol and BirA*-catalysed biotinylation.

Several recent studies have used BirA* to identify novel proteins in apicomplexan organelles (*Boucher et al., 2018*; *Chen et al., 2015*; *Kehrer et al., 2016*; *Nadipuram et al., 2016*). Our study is the first time APEX has been used for proteomic applications in apicomplexan parasites, and we suggest that this approach will prove useful in elucidating the proteomes of other apicomplexan organelles (e.g. the apicoplast) or organellar sub-compartments (e.g. the mitochondrial intermembrane space), either by itself or in combination with BirA* approaches. Our study suggests that APEX and BirA* are powerful tools in determining organellar proteomes in apicomplexans.

Our phylogenetic comparisons suggest that a large number of proteins in the *T. gondii* mitochondrion have homologs in *P. falciparum* (*Figure 4A*; *Supplementary file 1*). This is consistent with the presence of similar mitochondrial processes and biochemical pathways in these organisms (*Seeber et al., 2008*). The lower value in *C. parvum* is consistent with the reduced function of the mitosome of this organism (*Mogi and Kita, 2010*). The shared mitochondrial/mitosomal proteome of *T. gondii* and *C. parvum* likely consists of proteins involved in pathways shared between these organisms, such as Fe-S cluster synthesis. Notably, we identified homologs to 83% of *T. gondii* mitochondrial proteins in the chromerid *V. brassicaformis* (*Figure 4A*; *Supplementary file 1*). This mirrors recent findings that several 'derived' features of apicomplexan mitochondria, including the repurposing of the BCKDH, the loss of Complex I, and the acquisition of novel TCA cycle enzymes, were already present in the free-living, autotrophic common ancestor of myzozoans (dinoflagellates, chromerids and apicomplexans; *Danne et al., 2013*; *Jacot et al., 2016*). Notably, most of the ApiCox proteins we identified in the *T. gondii* COX complex had homologs in chromerids (*Table 1*), indicating that even conserved mitochondrial processes in this group of organisms have a considerable

degree of novelty. Our findings mirror those of two recent studies that examined the mitochondrial ATP synthase complex of *T. gondii*, which also contains many subunits that are conserved between apicomplexans and their nearest free-living relatives (*Huet et al., 2018*; *Salunke et al., 2018*). Together, these data indicate that much of the mitochondrial biology in *T. gondii* was present in the free-living ancestor that they share with chromerids.

Many (~40 – 50%) mitochondrial proteins in *T. gondii* lack apparent orthologs in animals and other eukaryotes (*Figure 4B–C*). Surprisingly, proteins found only in coccidians, or restricted to apicomplexans and chromerids, were just as likely to be important for parasite growth as proteins conserved across eukaryotic evolution (*Figure 4D*). This suggests that many derived or unique features of the *T. gondii* mitochondrion (and apicomplexan mitochondria more generally) are critical for parasite survival. This is in contrast to the general proteome of *T. gondii*, where proteins with a restricted phylogenetic distribution are typically less important for parasite survival (*Sidik et al., 2016*).

To understand the functions of apicomplexan-specific mitochondrial proteins that are important for parasite growth, we have commenced a project to characterize these proteins. In this study, we describe the characterisation of *Tg*ApiCox25, demonstrating that this is a component of the *T. gondii* COX complex. *Tg*ApiCox25 is important for parasite growth and mitochondrial oxygen consumption (*Figure 5*). Knockdown of *Tg*ApiCox25 also leads to defects in the integrity of the COX complex. In particular, *Tg*ApiCox25 knockdown leads to a depletion of *Tg*Cox2a abundance, and also results in the appearance of a smaller,~400 kDa *Tg*Cox2a-containing complex (*Figure 7E–F*). These data imply an important role for *Tg*ApiCox25 in the assembly and/or stability of the COX complex. It remains unclear whether loss of *Tg*ApiCox25 leads to loss of a ~ 200 kDa module from the complex, or whether *Tg*ApiCox25 knockdown leads to defects in COX assembly, with the ~400 kDa complex representing an assembly intermediate. Regardless, loss of *Tg*ApiCox25 results in defects in the abundance and integrity of the parasite COX complex, which likely explains the defects in mOCR observed upon *Tg*ApiCox25 knockdown. Future studies will examine the functional role of *Tg*ApiCox25 in the COX complex of the parasite.

Our analysis of the parasite COX complex revealed that it is approximately 600 kDa in mass (*Figure 6A*), which is larger than the equivalent complex in yeast (200 kDa), mammals (200 kDa) and plants (220/300 kDa) (*Eubel et al., 2003*; *Lenaz and Genova, 2010*; *Maréchal et al., 2012*; *Schägger and Pfeiffer, 2000*; *Tsukihara et al., 1996*). It is conceivable that the *T. gondii* COX complex exists in a multimeric form (e.g. a homodimer or homotrimer), and/or that it contains subunits not present in the complex of these other organisms that inflate its mass. Another possibility is that *T. gondii* COX exists in a 'super-complex' with other respiratory complexes, as has been observed in other systems (*Eubel et al., 2003*; *Schägger and Pfeiffer, 2000*). Although we cannot rule out that respiratory chain supercomplexes exist in *T. gondii*, the 600 kDa complex is probably not a super-complex, since (i) we do not observe enrichment of canonical components of other respiratory chain proteins in our proteomic analysis of the complex, (ii) we solubilised the cytochrome *c* oxidase complex with Triton X-100 detergent, rather than digitonin (which is typically used to solubilize super-complexes in other organisms; *Schägger and Pfeiffer, 2000*), and (iii) the molecular masses of other respiratory chain complexes in *T. gondii* do not correspond in size to COX (GvD, unpublished observations).

We identified 15 proteins in the *T. gondii* COX complex. In addition to these, it is likely that the complex contains *Tg*CoxI, a protein encoded on the mitochondrial genome of the parasite. 16 COX proteins in *T. gondii* is comparable in number to those found in the mammalian (13 proteins) and plant (10 or 12 proteins) complexes (*Eubel et al., 2003*; *Tsukihara et al., 1996*). Surprisingly, only a few *T. gondii* COX proteins have obvious homologs in eukaryotes outside the myzozoan lineage. Of note, 10 of the 11 ApiCox proteins we identified have homologs in chromerids (*Table 1*), while only *Tg*ApiCox13 has a clear homolog in ciliates (http://orthomcl.org), the eukaryotic lineage that is the sister taxon to myzozoans. This suggests either a high degree of novelty in the proteins that comprise the *T. gondii*/myzozoan COX complex, or that the sequences of ApiCox proteins have diverged to the extent that they are no longer easily recognisable by sequence comparisons. Notably, similarity searches that incorporate secondary structure information suggest that ApiCox25 and ApiCox23 may have homology to Cox6a and Cox4, respectively, from animals. A priority in the field is to establish the structure of the COX complex in myzozoans, which will reveal whether ApiCox proteins have structural equivalents in other eukaryotes. Regardless of their degree of novelty, our observations are consistent with other observations that suggest a considerable divergence in the

mitochondrial biology of organisms from the myzozoan lineage compared to other eukaryotes (*Danne et al., 2013*; *Huet et al., 2018*; *Jacot et al., 2016*; *Salunke et al., 2018*).

Our study highlights the divergence of mitochondrial proteomes across the eukaryotic domain of life. Future studies will be aimed at elucidating the function of other *T. gondii*- and myzozoan-specific mitochondrial proteins, since these proteins will provide insights into unique functions of the organelle. Given their importance to parasite growth, and their absence (or divergence) from the host organisms that they infect, these *T. gondii* and apicomplexan-specific proteins are prime drug targets.

# Materials and methods

## Key resources table

| Reagent type (species) or resource | Designation | Source or reference | Identifiers | Additional information |
|---|---|---|---|---|
| Strain, strain background (*Toxoplasma gondii*) | RHΔ*hxgprt* | PMID: 8662859 | | Parental strain for mtAPEX and mtBirA* strains |
| Strain, strain background (*T. gondii*) | mtAPEX-cmyc in RHΔ*hxgprt* | This paper | | mtAPEX-cmyc-expressing *T. gondii* |
| Strain, strain background (*T. gondii*) | mtBirA*-cmyc in RHΔ*hxgprt* | This paper | | mtBirA*-cmyc-expressing *T. gondii* |
| Strain, strain background (*T. gondii*) | TATi/Δ*ku80* | PMID: 22144892 | | Parental for 3' HA tag integration strains and r*Tg*ApiCox25 strain and derivatives thereof |
| Strain, strain background (*T. gondii*) | TGGT1_209420 HA in TATi/Δ*ku80* | This paper | | Gene contains integrated 3' HA tag |
| Strain, strain background (*T. gondii*) | TGGT1_209870 HA in TATi/Δ*ku80* | This paper | | Gene contains integrated 3' HA tag |
| Strain, strain background (*T. gondii*) | TGGT1_210730 HA in TATi/Δ*ku80* | This paper | | Gene contains integrated 3' HA tag |
| Strain, strain background (*T. gondii*) | TGGT1_213420 HA in TATi/Δ*ku80* | This paper | | Gene contains integrated 3' HA tag |
| Strain, strain background (*T. gondii*) | TGGT1_217020 HA in TATi/Δ*ku80* | This paper | | Gene contains integrated 3' HA tag |
| Strain, strain background (*T. gondii*) | TGGT1_220100 HA in TATi/Δ*ku80* | This paper | | Gene contains integrated 3' HA tag |
| Strain, strain background (*T. gondii*) | TGGT1_223500 HA in TATi/Δ*ku80* | This paper | | Gene contains integrated 3' HA tag |
| Strain, strain background (*T. gondii*) | TGGT1_229620 HA in TATi/Δ*ku80* | This paper | | Gene contains integrated 3' HA tag |
| Strain, strain background (*T. gondii*) | TGGT1_232600 HA in TATi/Δ*ku80* | This paper | | Gene contains integrated 3' HA tag |
| Strain, strain background (*T. gondii*) | TGGT1_248600 HA in TATi/Δ*ku80* | This paper | | Gene contains integrated 3' HA tag |

*Continued on next page*

*Continued*

| Reagent type (species) or resource | Designation | Source or reference | Identifiers | Additional information |
|---|---|---|---|---|
| Strain, strain background (*T. gondii*) | TGGT1_258100 HA in TATi/Δ*ku80* | This paper | | Gene contains integrated 3' HA tag |
| Strain, strain background (*T. gondii*) | TGGT1_259710 HA in TATi/Δ*ku80* | This paper | | Gene contains integrated 3' HA tag |
| Strain, strain background (*T. gondii*) | TGGT1_260840 HA in TATi/Δ*ku80* | This paper | | Gene contains integrated 3' HA tag |
| Strain, strain background (*T. gondii*) | TGGT1_263080 HA in TATi/Δ*ku80* | This paper | | Gene contains integrated 3' HA tag |
| Strain, strain background (*T. gondii*) | TGGT1_263400 HA in TATi/Δ*ku80* | This paper | | Gene contains integrated 3' HA tag |
| Strain, strain background (*T. gondii*) | TGGT1_263840 HA in TATi/Δ*ku80* | This paper | | Gene contains integrated 3' HA tag |
| Strain, strain background (*T. gondii*) | TGGT1_265360 HA in TATi/Δ*ku80* | This paper | | Gene contains integrated 3' HA tag |
| Strain, strain background (*T. gondii*) | TGGT1_264040 HA in TATi/Δ*ku80* | This paper | | Gene contains integrated 3' HA tag; termed *Tg*ApiCox25-HA strain |
| Strain, strain background (*T. gondii*) | TGGT1_275650 HA in TATi/Δ*ku80* | This paper | | Gene contains integrated 3' HA tag |
| Strain, strain background (*T. gondii*) | TGGT1_278250 HA in TATi/Δ*ku80* | This paper | | Gene contains integrated 3' HA tag |
| Strain, strain background (*T. gondii*) | TGGT1_278720 HA in TATi/Δ*ku80* | This paper | | Gene contains integrated 3' HA tag |
| Strain, strain background (*T. gondii*) | TGGT1_286120 HA in TATi/Δ*ku80* | This paper | | Gene contains integrated 3' HA tag |
| Strain, strain background (*T. gondii*) | TGGT1_290460 HA in TATi/Δ*ku80* | This paper | | Gene contains integrated 3' HA tag |
| Strain, strain background (*T. gondii*) | TGGT1_297810 HA in TATi/Δ*ku80* | This paper | | Gene contains integrated 3' HA tag; termed TgApiCox30-HA strain |
| Strain, strain background (*T. gondii*) | TGGT1_300030 HA in TATi/Δ*ku80* | This paper | | Gene contains integrated 3' HA tag |
| Strain, strain background (*T. gondii*) | TGGT1_316900 HA in TATi/Δ*ku80* | This paper | | Gene contains integrated 3' HA tag |
| Strain, strain background (*T. gondii*) | TGGT1_318400 HA in TATi/Δ*ku80* | This paper | | Gene contains integrated 3' HA tag |
| Strain, strain background (*T. gondii*) | r*Tg*ApiCox25 in TATi/Δ*ku80* | This paper | | ATc-regulated promoter driving *Tg*ApiCox25 expression |
| Strain, strain background (*T. gondii*) | r*Tg*ApiCox25-HA | This paper | | Regulatable *Tg*ApiCox25 gene with integrated 3' HA tag |
| Strain, strain background (*T. gondii*) | c*Tg*ApiCox25-HA/ r*Tg*ApiCox25 | This paper | | r*Tg*ApiCox25 strain complemented with constitutive *Tg*ApiCox25-HA |

*Continued on next page*

*Continued*

| Reagent type (species) or resource | Designation | Source or reference | Identifiers | Additional information |
|---|---|---|---|---|
| Strain, strain background (*T. gondii*) | *Tg*Cox2A-FLAG in *Tg*ApiCox25-HA | This paper | | Integrated 3' FLAG tag in *Tg*Cox2a locus of *Tg*ApiCox25-HA strain |
| Strain, strain background (*T. gondii*) | *Tg*Cox2A-FLAG in r*Tg*ApiCox25-HA | This paper | | Integrated 3' FLAG tag in *Tg*Cox2a locus of r*Tg*ApiCox25-HA strain |
| Strain, strain background (*T. gondii*) | *Tg*ApiCox25-FLAG in *Tg*ApiCox30-HA | This paper | | Integrated 3' FLAG tag in *Tg*ApiCox25 locus of *Tg*ApiCox30-HA strain |
| Cell line (*Homo sapians*) | Human Foreskin Fibroblasts | Gift from Holger Schülter, Peter MacCallum Cancer Centre | | |
| Antibody | mouse anti-cmyc | Santa Cruz | Clone 9E10 | (1:200 to 1:500) |
| Antibody | rat anti-HA | Sigma | Clone 3F10 | (1:200 to 1:1,000) |
| Antibody | mouse anti-FLAG | Sigma | Clone M2 | (1:500 to 1:2,000) |
| Antibody | rabbit anti-AtpB | Agrisera | cat #: AS05 085 | (1:500) |
| Antibody | rabbit anti-*Tg*Tom40 | PMID: 27458014 | | (1:2,000) |
| Antibody | rabbit anti-*Tg*CytC | This paper | | Peptide antibody made against residues 1–14 (MSRAEPDVQVPSGD) of *T. gondii* cytochrome *c* (TGGT1_219750) (1:500) |
| Antibody | rabbit anti-Hsp60 | PMID: 15279947 | | (1:1,000) |
| Antibody | goat anti-mouse Alexa Fluor 488 | Life Technologies | cat #: A11029 | (1:500) |
| Antibody | goat anti-rat Alexa Fluor 488 | Life Technologies | cat #: A11006 | (1:100 to 1:500) |
| Antibody | goat anti-rat CF 488A | Sigma | cat #: SAB4600046 | (1:100 to 1:500) |
| Antibody | goat anti-rabbit Alexa Fluor 546 | Life Technologies | cat #: A11035 | (1:500) |
| Antibody | goat anti-mouse HRP-conjugated | Santa Cruz | cat #: sc-2005 | (1:5,000) |
| Antibody | goat anti-rat HRP-conjugated | Santa Cruz | cat #: sc-2006 | (1:5,000) |
| Antibody | goat anti-rabbit HRP-conjugated | Santa Cruz | cat #: sc-2004 | (1:5,000) |
| Antibody | anti-mouse HRP-conjugated TrueBlot Ultra | eBioscience | cat #: 18-8817-31 | (1:5,000) |
| Antibody | anti-FLAG M2 affinity gel | Sigma | cat #: A2220 | |
| Antibody | anti-HA affinity matrix | Sigma | cat #: 11815016001 | |
| Peptide, recombinant protein | Avidin, Oregon Green -conjugated | Life Technologies | cat #: A6374 | (1:1,000) |
| Peptide, recombinant protein | NeutrAvidin, HRP-conjugated | Life Technologies | cat #: A2664 | (1:10,000) |
| Peptide, recombinant protein | Streptavidin magnetic beads | Thermo Scientific | cat #: PIE88817 | |
| Recombinant DNA reagent | pcDNA3-mito-APEX | PMID: 23086203 | Addgene cat # 42607 | |

*Continued on next page*

*Continued*

| Reagent type (species) or resource | Designation | Source or reference | Identifiers | Additional information |
|---|---|---|---|---|
| Recombinant DNA reagent | pBirA*−3XhA-LIC-DHFR | PMID: 25691595 | | |
| Recombinant DNA reagent | pSAG1::Cas9-U6::sgUPRT | PMID: 24825012 | Addgene cat # 54467 | |
| Recombinant DNA reagent | pUgCTH3 | PMID: 28205520 | | |
| Recombinant DNA reagent | mtAPEX-cmyc in pBTM3 | This paper | | *T. gondii* expression vector encoding mitochondrially targeted APEX |
| Recombinant DNA reagent | mtBirA*-cmyc in pBTM3 | This paper | | *T. gondii* expression vector encoding mitochondrially targeted APEX |
| Recombinant DNA reagent | TGGT1_209420 HA in pgCH | This paper | | *T. gondii* 3' replacment vector that introduces 1xHA tag into target gene |
| Recombinant DNA reagent | TGGT1_209870 HA in pgCH | This paper | | *T. gondii* 3' replacment vector that introduces 1xHA tag into target gene |
| Recombinant DNA reagent | TGGT1_210730 HA in pgCH | This paper | | *T. gondii* 3' replacment vector that introduces 1xHA tag into target gene |
| Recombinant DNA reagent | TGGT1_213420 HA in pgCH | This paper | | *T. gondii* 3' replacment vector that introduces 1xHA tag into target gene |
| Recombinant DNA reagent | TGGT1_217020 HA in pgCH | This paper | | *T. gondii* 3' replacment vector that introduces 1xHA tag into target gene |
| Recombinant DNA reagent | TGGT1_220100 HA in pgCH | This paper | | *T. gondii* 3' replacment vector that introduces 1xHA tag into target gene |
| Recombinant DNA reagent | TGGT1_223500 HA in pgCH | This paper | | *T. gondii* 3' replacment vector that introduces 1xHA tag into target gene |
| Recombinant DNA reagent | TGGT1_229620 HA in pgCH | This paper | | *T. gondii* 3' replacment vector that introduces 1xHA tag into target gene |
| Recombinant DNA reagent | TGGT1_232600 HA in pgCH | This paper | | *T. gondii* 3' replacment vector that introduces 1xHA tag into target gene |
| Recombinant DNA reagent | TGGT1_248600 HA in pgCH | This paper | | *T. gondii* 3' replacment vector that introduces 1xHA tag into target gene |
| Recombinant DNA reagent | TGGT1_258100 HA in pgCH | This paper | | *T. gondii* 3' replacment vector that introduces 1xHA tag into target gene |
| Recombinant DNA reagent | TGGT1_259710 HA in pgCH | This paper | | *T. gondii* 3' replacment vector that introduces 1xHA tag into target gene |
| Recombinant DNA reagent | TGGT1_260840 HA in pgCH | This paper | | *T. gondii* 3' replacment vector that introduces 1xHA tag into target gene |
| Recombinant DNA reagent | TGGT1_263080 HA in pgCH | This paper | | *T. gondii* 3' replacment vector that introduces 1xHA tag into target gene |

*Continued on next page*

*Continued*

| Reagent type (species) or resource | Designation | Source or reference | Identifiers | Additional information |
|---|---|---|---|---|
| Recombinant DNA reagent | TGGT1_263400 HA in pgCH | This paper | | *T. gondii* 3' replacment vector that introduces 1xHA tag into target gene |
| Recombinant DNA reagent | TGGT1_263840 HA in pgCH | This paper | | *T. gondii* 3' replacment vector that introduces 1xHA tag into target gene |
| Recombinant DNA reagent | TGGT1_265360 HA in pgCH | This paper | | *T. gondii* 3' replacment vector that introduces 1xHA tag into target gene |
| Recombinant DNA reagent | TGGT1_264040 HA in pgCH | This paper | | *T. gondii* 3' replacment vector that introduces 1xHA tag into target gene (*Tg*ApiCox25) |
| Recombinant DNA reagent | TGGT1_275650 HA in pgCH | This paper | | *T. gondii* 3' replacment vector that introduces 1xHA tag into target gene |
| Recombinant DNA reagent | TGGT1_278250 HA in pgCH | This paper | | *T. gondii* 3' replacment vector that introduces 1xHA tag into target gene |
| Recombinant DNA reagent | TGGT1_278720 HA in pgCH | This paper | | *T. gondii* 3' replacment vector that introduces 1xHA tag into target gene |
| Recombinant DNA reagent | TGGT1_286120 HA in pgCH | This paper | | *T. gondii* 3' replacment vector that introduces 1xHA tag into target gene |
| Recombinant DNA reagent | TGGT1_290460 HA in pgCH | This paper | | *T. gondii* 3' replacment vector that introduces 1xHA tag into target gene |
| Recombinant DNA reagent | TGGT1_297810 HA in pgCH | This paper | | T. gondii 3' replacment vector that introduces 1xHA tag into target gene (*Tg*ApiCox30) |
| Recombinant DNA reagent | TGGT1_300030 HA in pgCH | This paper | | *T. gondii* 3' replacment vector that introduces 1xHA tag into target gene |
| Recombinant DNA reagent | TGGT1_316900 HA in pgCH | This paper | | *T. gondii* 3' replacment vector that introduces 1xHA tag into target gene |
| Recombinant DNA reagent | TGGT1_318400 HA in pgCH | This paper | | *T. gondii* 3' replacment vector that introduces 1xHA tag into target gene |
| Recombinant DNA reagent | *Tg*ApiCox25 5' sgRNA in pSAG1::Cas9-U6 | This paper | | pSAG1::Cas9-U6 vector expressing sgRNA that targets 5' region of *Tg*ApiCox25 |
| Recombinant DNA reagent | *Tg*ApiCox25 3' sgRNA in pSAG1::Cas9-U6 | This paper | | pSAG1::Cas9-U6 vector expressing sgRNA that targets 3' region of *Tg*ApiCox25 |
| Recombinant DNA reagent | *Tg*Cox2a 3' sgRNA in pSAG1::Cas9-U6 | This paper | | pSAG1::Cas9-U6 vector expressing sgRNA that targets 3' region of *Tg*Cox2a |
| Recombinant DNA reagent | *Tg*ApiCox25 in pUgCTH3 | This paper | | Vector that expresses *Tg*ApiCox25-HA from the constitutive α-tubulin promoter |
| Software, algorithm | Mitochondrial Matrix Quantitative Proteome search tool | This paper | https://bit.ly/2FySSmU | Link to region of ToxoDB website containing the *T. gondii* mitochondrial proteome search tool |

## Parasite culture

*T. gondii* parasites were passaged in human foreskin fibroblasts (HFF) sourced from the Peter Mac-Callum Cancer Centre, and were verified mycoplasma-free. Parasites were cultured in Dulbecco's Modified Eagle's Medium, supplemented with 1% (v/v) fetal bovine serum and antibiotics. Where appropriate, ATc was added to a final concentration of 0.5 µg/ml. Plaque assays were performed as described previously (*van Dooren et al., 2008*), with plaque sizes measured using ImageJ.

## Plasmid construction and parasite transfection

To generate a *T. gondii* strain expressing mitochondrial matrix-targeted APEX, we amplified the coding sequence of monomeric APEX using the primers APEX fwd and APEX rvs (*Supplementary file 4*), and the vector pcDNA3-mito-APEX as template (a gift from Alice Ting; Addgene plasmid # 42607; *Martell et al., 2012*). The resultant PCR product was digested with *Avr*II and *Nde*I and ligated into equivalent sites of the vector Hsp60$_L$-mDHFR in pBTM$_3$ (*Figure 1—figure supplement 1*; *van Dooren et al., 2016*). This fuses the mitochondrial targeting sequence of *Tg*Hsp60 to 3x c-myc-tagged APEX. The resultant vector was transfected into RHΔ*hxgprt* strain *T. gondii* parasites, and selected on phleomycin as described (*Messina et al., 1995*). To generate a *T. gondii* strain expressing a mitochondrial matrix-targeted BirA*, we amplified the coding sequence of BirA* using the primers BirA* fwd and BirA* rvs (*Supplementary file 4*), and the vector pBir-A*−3XHA-LIC-DHFR as template (*Chen et al., 2015*; a gift from Peter Bradley, University of California Los Angeles). The resultant product was digested with *Bgl*II and *Xba*I and ligated into the *Bgl*II and *Avr*II sites of pgCM3 (GvD, unpublished). The BirA*−3x c-myc cassette of the resultant vector was digested with *Avr*II and *Not*I and ligated into the equivalent sites of the Hsp60$_L$-mDHFR in pBTM$_3$ vector. This fuses the mitochondrial targeting sequence of *Tg*Hsp60 to 3x c-myc-tagged BirA*. The resultant vector was transfected into RHΔ*hxgprt* strain *T. gondii* parasites, and selected on phleomycin.

For localizing candidate mitochondrial proteins from the proteome (including *Tg*ApiCox25 and *Tg*ApiCox30), we numbered proteins in the dataset and used a random number generator to select proteins from the list to localise. We excluded any proteins for which the localisation was known or suspected. We then amplified the 3' region of 27 target genes using the primers listed in *Supplementary file 4*. We digested resultant PCR products with enzymes suitable for subsequent ligation into the *Spe*I, *Bgl*II and/or *Avr*II sites of the vector pgCH (*Figure 3—figure supplement 1*; *Rajendran et al., 2017*), as outlined in *Supplementary file 4*. The resulting vector was linearized with an enzyme that cut once in the flanking sequence, transfected into TATi/Δ*ku80* strain parasites (*Sheiner et al., 2011*), and selected on chloramphenicol as described (*Striepen and Soldati, 2007*). The resultant parasite strains have a 1xHA tag fused to the 3' end of the open reading frame of the target gene, enabling subsequent localisation of the target protein by immunofluorescence assays.

To introduce an ATc-regulated promoter into the *Tg*ApiCox25 locus, we generated a vector expressing a sgRNA targeting the region around the start codon of *Tg*ApiCox25. To do this, we modified the vector pSAG1::Cas9-U6::sgUPRT (Addgene plasmid # 54467; *Shen et al., 2014*) using Q5 site-directed mutagenesis (New England Biolabs) as described previously (*Shen et al., 2014*). For site-directed mutagenesis, we used the primers ApiCox25 5' CRISPR fwd and the universal reverse primer (*Supplementary file 4*). We also PCR amplified the ATc-regulated promoter plus a 'spacer' region consisting of part of the *T. gondii* DHFR open reading frame and 3' UTR using the pPR2-HA3 vector (*Katris et al., 2014*) as template and the primers ApiCox25 5' reg fwd and ApiCox25 5' reg rvs (*Supplementary file 4*), which each contain 50 bp of sequence specific for the *Tg*ApiCox25 locus. The sgRNA expressing vector, which also expressed GFP-tagged Cas9, was co-transfected into TATi/Δ*ku80* strain parasites along with the ATc-regulatable promoter as described (*Striepen and Soldati, 2007*). GFP-positive parasites were selected and cloned 3 days following transfection. Clones were screened for successful integration of the ATc-regulatable promoter using the primers ApiCox25 5' screen fwd and ApiCox25 screen rvs, or t7s4 screen fwd and ApiCox25 screen rvs (*Supplementary file 4*).

To generate a vector that constitutively expressed *Tg*ApiCox25 for complementing the r*Tg*ApiCox25 mutant, we PCR amplified the *Tg*ApiCox25 open reading frame with the primers ApiCox25 comp fwd and ApiCox25 comp rvs (*Supplementary file 4*). We digested the resulting PCR product with *Bam*HI and *Avr*II and ligated this into the *Bgl*II and *Avr*II sites of the vector pUgCTH$_3$ (*Figure 5—*

*figure supplement 4*; *Rajendran et al., 2017*). The resulting vector was linearized with *Mfe*I, transfected into r*Tg*ApiCox25 parasites, and selected on chloramphenicol as described (*Striepen and Soldati, 2007*).

To FLAG tag the native locus of *Tg*Cox2a, we generated a vector expressing a sgRNA targeting the region around the stop codon of *Tg*Cox2a. To do this, we modified the pSAG1::Cas9-U6::sgUPRT vector using Q5 mutagenesis with the primer Cox2a 3'rep CRISPR fwd and the Universal Reverse primer (*Supplementary file 4*). We also amplified a FLAG tag containing 50 bp of flanking sequence either side of the *Tg*Cox2a stop codon, using the primers Cox2a 3' rep fwd and Cox2a 3' rep rvs, with FLAG tag template synthesized as a gBlock (IDT; *Supplementary file 4*). We co-transfected the plasmid and PCR product into *Tg*ApiCox25-HA or r*Tg*ApiCox25-HA strain parasites, selected GFP positive parasites by flow cytometry 3 days post-transfection, then screened for successful integrants using the primers Cox2a 3' screen fwd and Cox2a 3' screen rvs (*Supplementary file 4*).

To FLAG tag the native locus of *Tg*ApiCox25, we generated a vector expressing a sgRNA targeting the region around the stop codon of TgApiCox25, modifying the pSAG1::Cas9-U6::sgUPRT vector using Q5 mutagenesis with the primer ApiCox25 3'rep CRISPR fwd and the Universal Reverse primer (*Supplementary file 4*). We also amplified a FLAG tag containing 50 bp of flanking sequence either side of the *Tg*ApiCox25 stop codon, using the primers ApiCox25 3'rep fwd and ApiCox25 3'rep rvs, and the FLAG tag gBlock as template (*Supplementary file 4*). We co-transfected the plasmid and PCR product into *Tg*ApiCox30-HA strain parasites, selected GFP positive parasites by flow cytometry 3 days post-transfection, then screened for successful integrants using the primers ApiCox25 3'rep screen fwd and *Tg*ApiCox25 screen rvs (*Supplementary file 4*).

## Synthesis of biotin-phenol

Biotin phenol was synthesised as described previously (*Rhee et al., 2013*). 50 mg/ml biotin (Sigma) was slowly mixed with 1.1 equivalents of 2-(7-aza-1H-benzotriazole-1-yl)-1,1,3,3-tetramethyluronium (Sigma) and 3.0 equivalents of N,N-diisopropylethylamine (Sigma). The mixture was stirred for 10 min at room temperature, then 1.0 equivalent of tyramine (Sigma) was added slowly. The resulting solution was stirred overnight at room temperature. The synthesised biotin-phenol was purified from the unreacted material with a Reveleris flash chromatography system (Grace, MD, USA) and a C18-WP 4 g column, using an acetonitrile/water gradient. Eluting compounds were monitored with a UV detector (220 nm, 278 nm, and 350 nm) and an evaporative light scattering detector (ELSD) coupled to the flash chromatography system. The eluted biotin-phenol in acetonitrile/water mixture was freeze-dried and reconstituted in dimethyl sulfoxide at a final concentration of 200 mM. The purity of biotin-phenol was confirmed by ultra-high performance liquid chromatography (UHPLC, Dionex).

## Biotinylation approaches

For biotin-phenol labelling, freshly egressed wild type or mtAPEX-expressing parasites were resuspended in parasite growth medium. Biotin-phenol was added to the parasites at final concentration of 1 mM, and parasites were incubated at 37°C for 1 hr. Biotinylation was initiated by the addition of 1 mM $H_2O_2$ for 45 s, and halted by centrifuging cells at 12,000 g for 30 s. The medium was removed, and parasite cells were washed three times in quenching solution (10 mM sodium azide, 10 mM sodium ascorbate, and 5 mM Trolox in phosphate-buffered saline (PBS)), and once in PBS. Cell pellets were stored at −80°C until further processing.

For biotin-labelling, host cells were infected with wild type or mtBirA*-expressing parasites. Biotin was added to infected host cells ~ 48 hr after infection, to a final concentration of 1 mM. Infected host cells were cultured for a further 24 hr, during which time they naturally egressed from host cells. Parasites were harvested by centrifugation at 1500 g and washed 3 times in PBS. Cell pellets were stored at −80°C until further processing.

Biotin treated parasite pellets were lysed in radioimmunoprecipitation assay (RIPA) buffer (50 mM Tris, 150 mM NaCl, 0.1% (w/v) SDS, 0.5% (w/v) sodium deoxycholate, 1% (v/v) Triton X-100, Complete protease cocktail [Roche]) for 30 min on ice. Biotin-phenol treated parasite pellets were lysed in RIPA buffer containing quenching agents (10 mM sodium azide, 10 mM sodium ascorbate, and 5 mM Trolox). Three independent lysate pools of WT parasites treated with biotin-phenol and $H_2O_2$, mtAPEX-expressing parasites treated with biotin-phenol and $H_2O_2$, WT parasites treated with biotin,

and mtBirA*-expressing parasites treated with biotin were generated, and protein concentration was quantified using Bradford reagent (Sigma).

For enrichment of biotinylated proteins with streptavidin beads, 8 mg of protein from each pool was diluted with RIPA buffer to reach 1.8 ml total volume. 223 µl of streptavidin-conjugated magnetic beads (Thermo Scientific) per pool was dispensed in 2 ml microcentrifuge tubes. A magnabind magnet (Thermo Scientific) was used to separate beads from the buffer solution. Beads were washed three times in RIPA buffer, and incubated with the corresponding lysate pools for 1 hr at room temperature with gentle rotation. The beads were then washed three times in RIPA buffer, once with 2 M urea in 10 mM Tris-HCl pH 8.0, and a further three times in RIPA buffer. The resin-bound proteins were treated with reducing solution (10 mM DTT in 100 mM ammonium bicarbonate) for 30 min at 56°C. The reducing solution was removed, and replaced with alkylation solution (55 mM iodoacetamide in 100 mM ammonium bicarbonate) and incubated at room temperature in the dark for 45 min. The alkylation solution was then removed, and the beads were washed in 50 mM ammonium bicarbonate for 15 min with gentle agitation. The ammonium bicarbonate solution was removed, and samples treated with 20 ng/µl sequencing grade trypsin (Promega) overnight (18 hr) at 37°C. The next day, the supernatant was collected and beads were further treated with 10% (v/v) formic acid, and incubated for 15 min at 37°C.

The volume of peptide filtrates was reduced to ~12 µl in a centrifugal evaporator. Zip tips containing 0.6 µl C18 resin (Millipore) were washed with 10 µl methanol, and 10 µl 0.1% (v/v) formic acid. The digested peptides were loaded onto the tips by pipetting the peptide solutions 10 times. The tips were then washed with 10 µl 0.1% (v/v) formic acid twice, and the peptides were eluted into Chromacol 03-FISV vials with conical 300 µl inserts using 50% acetonitrile/0.1% formic acid (v/v), and dried in a vacuum concentrator.

MS analysis was performed on a Q-Exactive Classic mass spectrometer as previously described (*Delconte et al., 2016*). Raw files consisting of high-resolution MS/MS spectra were processed with MaxQuant (version 1.5.2.8) for feature detection and protein identification using the Andromeda search engine (*Cox et al., 2011*). Extracted peak lists were searched against the UniProtKB/SwissProt *Homo sapiens* and *Toxoplasma gondii* ME49 (ToxoDB-12.0) databases and a separate reverse decoy database to empirically assess the false discovery rate (FDR) using strict trypsin specificity allowing up to three missed cleavages. The minimum required peptide length was set to seven amino acids. *Modifications:* Carbamidomethylation of Cys was set as a fixed modification, while N-acetylation of proteins and oxidation of Met were set as variable modifications. The mass tolerance for precursor ions and fragment ions were 20 ppm and 0.5 Da, respectively. The 'match between runs' option in MaxQuant was used to transfer identifications made between runs on the basis of matching precursors with high mass accuracy (*Cox and Mann, 2008*). PSM and protein identifications were filtered using a target-decoy approach at a false discovery rate (FDR) of 1%.

## Quantitative proteomics pipeline

Statistically-relevant protein expression changes were identified using a custom pipeline as previously described (*Delconte et al., 2016*). Probability values were corrected for multiple testing using Benjamini–Hochberg method. Cut-off lines with the function $y = -\log_{10}(0.05) + c/(x-x_0)$ (*Keilhauer et al., 2015*) were introduced to identify significantly enriched proteins. c was set to 0.2 while $x_0$ was set to one representing proteins that are differentially expressed within 1 or two standard deviations.

## Bioinformatic analyses of data

Homologs of *T. gondii* mitochondrial proteome proteins were identified in the apicomplexan parasites *P. falciparum* (strain 3D7), *C. parvum* and *B. bovis*, and the chromerid *V. brassicaformis*, through reciprocal Basic Local Alignment Search Tool (BLAST) searches. *T. gondii* mitochondrial proteome proteins were used as query sequences in initial searches using target protein databases from relevant EuPathDB websites (http://PlasmoDB.org– *P. falciparum*; http://PiroplasmaDB.org – *Babesia bovis*; http://CryptoDB.org – *Cryptosporidium parvum* and *Vitrella brassicaformis*; *Aurrecoechea et al., 2013*). Hits from the initial BLAST search were queried in reciprocal BLAST searches against the *T. gondii* genome database (http://toxodb.org), regardless of the score or E-value obtained. Hits that returned the corresponding *T. gondii* protein that was originally searched

against were considered as a homolog. Expect (E) values obtained from the initial BLAST search and the reciprocal BLAST search were recorded (*Supplementary file 1*). Homologs of proteins identified in the purified ApiCox25 complex were identified using the profile hidden Markov model search tool HMMER (https://www.ebi.ac.uk/Tools/hmmer/).

Ortholog grouping for each protein was obtained from ToxoDB. Each ortholog group was assessed for the presence of ortholog proteins in other eukaryotic organisms based on the phyletic information available on http://OrthoMCL.org (*Chen et al., 2006*). Information of ortholog grouping for the entire genome of the chromerid *V. brassicaformis* were sourced from CryptoDB, and were subsequently compared to the ortholog groups identified for the *T. gondii* mitochondrial proteome. HHPRED predictions were performed using the MPI Bioinformatics Toolkit website (https://toolkit.tuebingen.mpg.de/#/tools/hhpred; *Zimmermann et al., 2018*).

The presence of predicted mitochondrial targeting peptides was assessed using MitoProt (*Claros and Vincens, 1996*), with the probability of export into the mitochondria recorded (*Supplementary file 1*). Metabolic pathway enrichment was assessed using the Metabolic Pathway search tool on ToxoDB, using a *p* value cut-off of <0.05.

## Immunofluorescence assays and microscopy

IFAs were performed as described previously (*van Dooren et al., 2008*). Primary antibodies used were mouse anti-c-myc (1:200 dilution; Santa Cruz clone 9E10), rat anti-HA (1:200 dilution; Sigma clone 3F10), and rabbit anti-Tom40 (1:2000 dilution; *van Dooren et al., 2016*). Secondary antibodies used were goat anti-mouse Alexa Fluor 488 (1:500 dilution; Life Technologies), goat anti-rat Alexa Fluor 488 (1:100 to 1:500 dilution; Life Technologies), goat anti-rat CF 488A (1:100 to 1:500 dilution; Sigma), and goat anti-rabbit Alexa Fluor 546 (1:500 dilution; Life Technologies). Biotinylation was performed as outlined for the proteomics, except that mtAPEX samples were incubated in $H_2O_2$ for 1 min. For visualizing biotinylated proteins, we used Oregon Green-conjugated avidin (1:1000 dilution; Life Technologies). Images were acquired on a DeltaVision Elite deconvolution microscope (GE Healthcare) fitted with a 100X UPlanSApo oil immersion objective lens (NA 1.40). Images were deconvolved and adjusted for contrast and brightness using SoftWoRx Suite 2.0 software, and subsequently processed using Adobe Illustrator.

## Immunoprecipitations

Immunoprecipitations were performed as described previously (*van Dooren et al., 2016*), except that parasite samples were solubilized in 1% (v/v) Triton X-100. HA-tagged proteins were purified using anti-HA affinity matrix (Sigma; rat anti-HA clone 3F10 antibodies) and FLAG-tagged proteins were purified using anti-FLAG M2 affinity gel (Sigma; mouse anti-FLAG clone M2 antibodies). For mass spectrometry sample preparation, anti-HA beads bound with HA-tagged protein complexes were frozen at −80°C for 1 hr, then eluted at 37°C in 0.2 M glycine containing 1% (v/v) Triton X-100 (pH 2.3). Samples were neutralized in ammonium bicarbonate, then extracted in chloroform:methanol as described (*Pankow et al., 2016*). After extraction, the pellets were dried and stored at −80°C before mass spectrometry analysis.

For the analysis, the protein pellets were dissolved in digestion buffer (8 M urea, 50 mM $NH_4HCO_3$, 10 mM dithiothreitol) and incubated at 25°C for 5 hr. Following incubation, iodoacetamide was added to a final concentration of 55 mM to alkylate thiol groups and incubated for 35 min at 20°C in the dark. The alkylated protein preparations were diluted to 1 M urea in 25 mM ammonium bicarbonate (pH 8.5) and sequencing-grade trypsin (Promega) was added to a ratio of 1:50 (w/w) to the samples. The reaction was incubated for 16 hr at 37°C. The digests were acidified with 1% (v/v) trifluoroacetic acid (TFA), dried in a SpeedVac centrifuge followed by a desalting step on SDB-XC StageTips (Empore, SDB-XC reversed-phase material, 3M, St. Paul, USA). Briefly, the digested proteins were resuspended in 100 μL of 1% (v/v) formic acid and centrifuged at 14 000 rpm for 2 min. The solid-phase extraction was performed according to (*Rappsilber et al., 2007*) with the following modifications: the membrane was conditioned with 50 μL of 80% (v/v) acetonitrile, 0.1% (w/v) TFA, and then washed with 50 μL of 0.1% TFA before the tryptic peptides were bound to the membrane. The bound peptides were eluted by 50 μL 80% (v/v) acetonitrile, 0.1% (w/v) TFA, and dried in a SpeedVac centrifuge.

Peptides reconstituted in 0.1% TFA and 2% acetonitrile (ACN) were loaded using a Thermo Scientific UltiMate 3000 RSLCnano system onto a trap column (C18 PepMap 300 µm ID ×2 cm trapping column, Thermo Fisher Scientific) at 15 µl/min for six minutes. The valve was then switched to allow the precolumn to be in line with the analytical column (Vydac MS C18, 3 µm, 300 Å and 75 µm ID ×25 cm, Grace Pty. Ltd.). The separation of peptides was performed at 300 nl/min at 45°C using a linear ACN gradient of buffer A (water with 0.1% formic acid, 2% ACN) and buffer B (water with 0.1% formic acid, 80% ACN), starting at 5% buffer B to 45% over 105 min, then 95% B for five minutes followed by an equilibration step of 15 min (water with 0.1% formic acid, 2% ACN). Data were collected on an Orbitrap Elite (Thermo Fisher Scientific) in Data Dependent Acquisition mode using m/z 300 – 1500 as MS scan range, CID MS/MS spectra were collected for the 10 most intense ions at a normalized collision energy of 35% and an isolation width of 2.0 m/z. Dynamic exclusion parameters were set as follows: repeat count 1, duration 90 s, the exclusion list size was set at 500 with early expiration disabled. Other instrument parameters for the Orbitrap were the following: MS scan at 120 000 resolution, maximum injection time 150 ms, AGC target $1 \times 10^6$ for a maximum injection time of 75 ms with AGT target of 5000.

The spectra obtained from the instrument were used to search against the UniProt TOXGV (*T. gondii*) database together with common contaminants using the Mascot search engine (Matrix Science Ltd., London, UK). Briefly, carbamidomethylation of cysteines was set as a fixed modification, acetylation of protein N-termini and methionine oxidation was included as variable modifications. Precursor mass tolerance was 10 ppm, product ions were searched at 0.5 Da tolerances, maximum of 2 missed trypsin cleavages, minimum peptide length defined at 6, maximum peptide length 144, and peptide spectral matches were validated using Percolator based on q-values at a 1% false discovery rate.

## SDS-PAGE, Blue Native-PAGE and immuno/affinity-blotting

SDS-PAGE and protein blotting were performed as described previously (*van Dooren et al., 2008*), except that membranes used for neutravidin blotting were blocked with 3% (w/v) bovine serum albumin (BSA). Blue native PAGE was performed using the NativePAGE system (Thermo Scientific) as described previously (*van Dooren et al., 2016*). Blots were probed with antibodies against mouse anti-c-myc (1:500 dilution; Santa Cruz clone 9E10), rabbit anti-Hsp60 (1:1000 dilution; *Tonkin et al., 2004*), rabbit anti-*T. gondii* cytochrome c (1:500 dilution; E.T. and G.v.D., unpublished), rat anti-HA (1:500 to 1:1000 dilution; Sigma clone 3F10), mouse anti-FLAG (1:500 to 1:2000 dilution; Sigma clone M2), and rabbit anti-AtpB (1:500; Agrisera, catalog number AS05 085). Horseradish peroxidase (HRP)-conjugated anti-mouse, anti-rat and anti-rabbit antibodies (Santa Cruz) were used at 1:5000 dilution. For probing for mouse antibodies on immunoprecipitation western blots, HRP-conjugated anti-mouse TrueBlot ULTRA antibodies (eBioscience) were used at 1:5000 dilution. Neutravidin-HRP (Life Technologies) was used to detect biotinylated proteins on membranes at 1:10,000 dilution.

## Seahorse XFe96 extracellular flux analysis

Wild type (TATi/Δku80) and rTgApiCox25 parasites were grown in the absence of ATc, in the presence of ATc for 1 – 3 days, or with 100 µM cycloheximide for 1 day. Parasites were filtered through a 3 µm polycarbonate filter and washed twice in Seahorse XF base medium (Agilent Technologies), supplemented with 1 mM L-glutamine and 5 mM D-glucose (supplemented base medium), before resuspension to $1.5 \times 10^7$ cells/ml in the supplemented base medium. 96-well Seahorse culture plates were coated with 3.5 µg/cm² of CellTak cell adhesive (Corning) according to the manufacturer's instructions. Briefly, 1 mg/ml CellTak was diluted 1:50 in 0.1 M sodium bicarbonate. 15 µl of CellTak solution was added to each well of a Seahorse cell culture plate and incubated at RT for 20 mins. The solution was removed and the plate washed twice in sterile water, before drying. 100 µl of the parasite suspensions ($1.5 \times 10^6$ parasites) were seeded into wells of the coated plate, and the plate was centrifuged at 50 g for 3 min. An additional 75 µL of supplemented base medium was added to each well following centrifugation. Parasites were kept at 37°C in a non-$CO_2$ incubator until the start of experiment. Parasite oxygen consumption rates (OCR) and extracellular acidification rates (ECAR) were measured using an Agilent Seahorse XFe96 Analyzer at 3 min intervals. To determine the maximal OCR, parasites were treated with 20 µM oligomycin A, B and C mix (Sigma) to inhibit ATP synthase, then subsequently treated with 1 µM carbonyl cyanide-4-(trifluoromethoxy)

phenylhydrazone (FCCP; Sigma). To determine the non-mitochondrial OCR, parasites were treated with 10 µM antimycin A (Sigma) and 1 µM atovaquone (the minimal concentration that preliminary experiments indicated is sufficient to maximally inhibit mitochondrial OCR). The mitochondrial OCR (mOCR) was calculated by subtracting the non-mitochondrial OCR from the basal and maximal OCR values. A minimum of 4 wells were used for background correction in each assay plate, and 4 – 5 technical replicates were used for each condition. Wells that yielded negative OCR values were excluded from the final analysis.

## Miscellaneous data analysis

XFe96 data were compiled using the Wave Desktop program. Analysis of parasite OCR and ECAR were performed using the R software environment (*Source code 1*). Linear mixed-effects models were fitted to the data, with error between plates and wells (i.e. between and within experiments) defined as the random effect, and the OCR and ECAR measurements in the different parasite strains (WT and r*Tg*ApiCox25) and the time since ATc-addition defined as the fixed effect. Data from the *Tg*ApiCox25-HA and *Tg*Tom40-HA co-immunopreciptation proteomics were analysed in the R software environment using the EdgeR package (*Source code 2*; *Robinson and Smyth, 2008*). Only proteins identified in both experimental conditions and each biological replicate were included in the final analysis. Graphing of the XFe96 and *Tg*ApiCox25 proteomic data were performed in Graph-Pad Prism v. 7.0.

## Data availability

The mitochondrial proteome data are available on individual gene pages on the ToxoDB website (http://toxodb.org), and a 'Mitochondrial Matrix Quantitative Proteome' search tool is available in the proteomics section of ToxoDB (https://bit.ly/2FySSmU). To use the search tool, first select the experiment that you want to query. 'Control APEX vs mito APEX' queries the APEX data, and 'Control BirA vs mito BirA' queries the BirA* data. Next select the direction of the query. To examine genes that are enriched in the mitochondrial matrix proteomes, select 'down-regulated' (i.e. proteins that are less abundant in control samples than in the mito APEX/BirA* samples). Next select the desired P value. For our analyses, we utilised a P value of $\leq 0.001$. Finally, select the desired fold difference. For our analyses, we utilised a $\log_2$ fold change value of $\leq -2.5$, which corresponds to a fold change between the experimental and control samples of $\geq 5.657$ (i.e. 5.657-fold down-regulated).

## Acknowledgements

We thank Peter Bradley (UCLA) and Alice Ting (Stanford) for sharing reagents, Michael Devoy and Harpreet Vohra (ANU) for assistance with flow cytometry, Michael Devoy (ANU) for assistance in establishing the Seahorse XFe96 assays, Teresa Neeman from the ANU Statistical Consulting Unit for assistance with data analysis, and Sebastian Lourido (Whitehead Institute) for comments on the manuscript. We are grateful to EuPathDB for providing numerous datasets and search tools, and in particular to Susanne Warrenfeltz for co-ordinating the integration of the mitochondrial matrix proteomic data into ToxoDB. This work was supported by a Discovery Grant and QEII fellowship from the Australian Research Council (ARC DP110103144) to GvD.

## Additional information

### Funding

| Funder | Grant reference number | Author |
| --- | --- | --- |
| Australian Research Council | DP110103144 | Giel G van Dooren |

The funders had no role in study design, data collection and interpretation, or the decision to submit the work for publication.

## Author contributions
Azadeh Seidi, Conceptualization, Data curation, Formal analysis, Validation, Investigation, Visualization, Methodology, Writing—original draft, Writing—review and editing; Linden S Muellner-Wong, Data curation, Formal analysis, Validation, Investigation, Visualization, Methodology, Writing—original draft, Writing—review and editing; Esther Rajendran, Conceptualization, Data curation, Formal analysis, Supervision, Validation, Investigation, Visualization, Methodology, Writing—original draft, Writing—review and editing; Edwin T Tjhin, Laura F Dagley, Data curation, Formal analysis, Investigation, Methodology, Writing—original draft, Writing—review and editing; Vincent YT Aw, Data curation, Formal analysis; Pierre Faou, Data curation, Formal analysis, Methodology; Andrew I Webb, Christopher J Tonkin, Resources, Supervision, Funding acquisition, Project administration, Writing—review and editing; Giel G van Dooren, Conceptualization, Resources, Data curation, Formal analysis, Supervision, Funding acquisition, Validation, Investigation, Visualization, Writing—original draft, Project administration, Writing—review and editing

## Author ORCIDs
Linden S Muellner-Wong (iD) https://orcid.org/0000-0003-0348-6408
Giel G van Dooren (iD) http://orcid.org/0000-0003-2455-9821

## Decision letter and Author response
Decision letter https://doi.org/10.7554/eLife.38131.030
Author response https://doi.org/10.7554/eLife.38131.031

# Additional files

## Supplementary files
• Supplementary file 1. List of genes encoding putative *T. gondii* mitochondrial proteins. Table 1. List of peptides identified in the APEX and BirA* proteomic analyses. Included are the ToxoDB accession numbers, the identified peptide, the experiment in which the peptide was identified, and the charge, m/z ratio, mass error, posterior error probability, score, delta score and intensity of each peptide from each experiment. Table 2. List of proteins identified in the RH control and mtAPEX proteomes, including the ToxoDB accession numbers, the $\log_2$ protein ratios, *p* value, and unique sequence counts. Table 3. List of proteins identified in the RH control and mtBirA* proteomes, including the ToxoDB accession numbers, the $\log_2$ protein ratios, *p* value, and unique sequence counts. Table 4. Summary of putative mitochondrial proteins identified in this study. The summary includes the ToxoDB accession numbers and annotation of proteins identified from the combined list (colums A and B), proteins identified in both lists (columns D and E), and proteins identified in the mtAPEX proteome (columns G and H) or mtBirA* (columns J and K) proteomes only. Proteins highlighted in green were demonstrated by this study to localise to the mitochondrion, while those highlighted in red did not localise to the mitochondrion. Table 5. Annotated protein list of the *T. gondii* mitochondrial proteome, noting the ToxoDB accession number of the corresponding gene, the protein annotation, mean phenotype score, molecular mass, number of transmembrane domains, amino acid sequence, MitoFates and MitoProt II prediction scores, the ortholog grouping, and the accession number of orthologous genes in *P. falciparum*, *C. parvum*, *B. bovis*, *V. brassicaformis*, and *S. cerevisiae* based on reciprocal BLAST searches. Homologs identified in *S. cerevisiae* were queried against the 'high confidence' mitochondrial proteome (Sc mito proteome) described in (*Morgenstern et al., 2017*). Table 6. Summary of the OrthoMCL analysis of the *T. gondii* mitochondrial proteome, depicting the gene annotation, mean phenotype score, and the relevant orthology grouping.
DOI: https://doi.org/10.7554/eLife.38131.019

• Supplementary file 2. Summary of metabolic pathway enrichment in the *T. gondii* mitochondrial proteome.
DOI: https://doi.org/10.7554/eLife.38131.020

• Supplementary file 3. Expected mitochondrial proteins and false negatives identified from the *T. gondii* mitochondrial proteome. List of proteins identified in the *T. gondii* mitochondrial proteome that previous studies have demonstrated or predicted to localize to the mitochondrion, and proteins

that previous studies have demonstrated do not localize to the mitochondrion. Included are the protein annotation, the process in which it functions, the proteome in which it was detected, and the ToxoDB gene ID. Note that natively biotinylated proteins (including the mitochondrially-localized pyruvate carboxylase; *Nitzsche et al., 2017*) were excluded from these analyses because of their biased (i.e. APEX and BirA*-independent) enrichment in control and experimental conditions. Color coding: green, predicted mitochondrial protein present in proteome; pink, predicted mitochondrial protein absent from proteome.
DOI: https://doi.org/10.7554/eLife.38131.021

• Supplementary file 4. List of primers and templates used in this study. Table 1. Primers and templates used in general cloning. Table 2. Primers used in 3' replacement localization studies. 3' fragments of target genes (ToxoDB gene ID) were amplified using the listed forward and reverse primers. The resulting PCR product was digested and ligated into the vector pgCH as outlined in the cloning strategy. The final vector was linearized with the indicated restriction enzyme before transfection.
DOI: https://doi.org/10.7554/eLife.38131.022

• Supplementary file 5. Table 1: List of proteins identified in the *Tg*ApiCox25 and *Tg*Tom40 immunoprecipitations. Included is a description of each identified protein, the UniProt accession number, the predicted molecular mass, the fold change, the normalized total precursor intensity for each biological replicate, and the Cox or ApiCox designation of the identified protein. Table 2: A list of the log fold change (logFC) and *p* values calculated for each protein identified in all replicates of the *Tg*ApiCox25 and *Tg*Tom40 immunoprecipitations following EdgeR analysis.
DOI: https://doi.org/10.7554/eLife.38131.023

• Source code 1. R script used in the analysis of the Seahorse XFe96 data.
DOI: https://doi.org/10.7554/eLife.38131.024

• Source code 2. R script used in the analysis of proteomic data from the *Tg*ApiCox25 and *Tg*Tom40 immunoprecipitations.
DOI: https://doi.org/10.7554/eLife.38131.025

• Transparent reporting form
DOI: https://doi.org/10.7554/eLife.38131.026

## Data availability

Mitochondrial proteomics data are available in on the ToxoDB website (http://toxodb.org).

The following dataset was generated:

| Author(s) | Year | Dataset title | Dataset URL | Database, license, and accessibility information |
|---|---|---|---|---|
| Giel G van Dooren | 2018 | Mitochondrial Matrix Quantitative Proteome | https://bit.ly/2FySSmU | Publicly available at ToxoDB (https://toxodb.org). |

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
