## [Decision Letter]

Thank you for submitting your article "Identification of a divergent cytochrome *c* oxidase complex in the mitochondrial proteome of *Toxoplasma gondii*" for consideration by *eLife*. Your article has been reviewed by three peer reviewers, and the evaluation has been overseen by a Reviewing Editor and Anna Akhmanova as the Senior Editor. The following individuals involved in review of your submission have agreed to reveal their identity: Sean Prigge (Reviewer #2); Akhil Vaidya (Reviewer #3).

The reviewers have discussed the reviews with one another and the Reviewing Editor has drafted this decision to help you prepare a revised submission.

Summary:

Seidi et al. provides a detailed proteomic view of *Toxoplasma gondii* mitochondrion by employing two different methods of proximity biotinylation followed by LC-MS/MS analysis. This is an important contribution laying a foundation for a more thorough understanding of mitochondrial functions in apicomplexan parasites. Among the 421 mitochondrial proteins identified here are many that appear to be unique to the myzozoan unicellular eukaryotes. Functional characterization of these would be a fertile endeavor that would reveal unique aspects of mitochondrial physiology in deep-branching eukaryotes. This would have significant implications for our understanding mitochondrial evolutions as well as providing potential avenues for therapeutic approaches for parasitic infections. To wit, authors characterize one such clade-specific protein, showing it to be a component of cytochrome *c* oxidase complex and use it to derive a fuller view of this complex that now appears to possess many additional unique subunits.

The authors build upon their proteomic analysis by thoroughly characterizing Cox25, a mitochondrial protein found only in apicomplexan lineages. Knockdown demonstrated that Cox25 is essential for parasite growth and reduces O2 consumption, consistent with a critical role in oxidative phosphorylation. The paper is well written, thorough and experimentally and statistically rigorous.

Essential revisions:

1) The comparison of the data sets suggests that the BirA method is significantly more error-prone than the APEX method. Indeed, only 1 of the 6 proteins experimentally localized from the APEX-only set was found to be a false positive while 4 of the 9 from the BirA-only data set were false positives. The authors could make a more forceful statement in this regard, perhaps supported by an analysis of whether the APEX-only dataset is enriched with predicted mito proteins compared to the BirA-only dataset.

Also in this context it is not evident to the reader that the 2 reported mitoproteomes come from two different parasite populations – extracellular (APEX) vs. intracellular (BioID) cultures. Importantly intracellular and extracellular parasites reflect two different biological states with marked changes in mRNA expression when transitioning from one to the other state (Gaji et al., Mol Microbiol. 2011). Parasites show also changes in metabolism when kept extracellular for more than an hour (Lin, S. et al., Int J Parasitol. 2011). Pyruvate carboxylase is known to be transcribed but not translated under some conditions (Nitzsche et al., 2017). In consequence, time scale during which parasites were kept extracellular under APEX labeling/washing/handling conditions could well influence the results and showed be discussed.

2) The authors make little use of a database like Mitominer and mentioned analyses in the respective publication (Smith, A. C. and Robinson, A. J. MitoMiner. Nucleic Acids Res 2016). This includes the usage of more than one targeting algorithm like iPSORT, TargetP, MitoFates etc. and compare their outcomes, as it is common use. It will make the statements in the first paragraph of the subsection “Bioinformatic characterisation of the mitochondrial proteome” more convincing. Also, using MiotMiner a more comprehensive comparison to other proteomes should be easy to perform and be informative with regard to a statement like "Our proteome identified most of the proteins 'expected' to localize to the *T. gondii* mitochondrion suggesting a high level of coverage".

3) Subsection “Spatially-restricted biotinylation of mitochondrial matrix proteins”, third paragraph: it is surprising that no mitochondrial labeling in absence of biotin was observed in Figure 1E, given that pyruvate carboxylase is present in mitos (TGME49_284190; with >350 spectra in the author's dataset at ToxoDB) and others observe staining in IFAs with streptavidin (e.g. Jelenska 2001). Please comment. Also, this protein is *not* part of Supplementary file 3 of known mito proteins, although its mitochondrial localization has been reported (Nitzsche et al., 2017). The latter work also shows PEPCK1 as mitochondrial, whereas here it is indicated as false positive (based on an older publication, Fleige IJP 2008). A more rigorous discussion and evaluation of this data should be included.

4) The authors did not detect any of the 3 proteins encoded by the mitochondrial genome, which is a bit curious. Could it be that we still do not have an authentic mtDNA sequence for Toxoplasma, which could affect proteomic detection? The authors should elaborate on the possible reasons why the products of the mitochondrial resident genes remain undetected.

---

## [Author Response]

Essential revisions:1) The comparison of the data sets suggests that the BirA method is significantly more error-prone than the APEX method. Indeed, only 1 of the 6 proteins experimentally localized from the APEX-only set was found to be a false positive while 4 of the 9 from the BirA-only data set were false positives. The authors could make a more forceful statement in this regard, perhaps supported by an analysis of whether the APEX-only dataset is enriched with predicted mito proteins compared to the BirA-only dataset.

We agree with the reviewers. We did make this point in the original manuscript (most notably in the section of the Discussion that reads: ‘Our findings suggest a low false positive rate in the proteins identified from the APEX proteome alone (~5%), but a higher false positive rate in the mtBirA* proteome (~20%). Based on these analyses, we consider the 213 proteins from the mtAPEX and shared mtAPEX/mtBirA* proteomes to be ‘likely’ mitochondrial proteins, and the 208 proteins found in the mtBirA* proteome alone to be ‘possible’ mitochondrial proteins.’), but have now extended our analyses and Discussion to address this further.

We have modified Supplementary file 3 to include a column (titled ‘Proteome’) in the ‘Expected Mitochondrial Proteins” section, that indicates which proteome (APEX alone, BirA alone, or both) each protein was identified. Our initial analysis identified 97 out of 111 ‘expected’ mitochondrial proteins in the mitochondrial proteomes. Of these 97 ‘true positives’, 52 (54%) were present in both proteomes, 18 (19% ) were present in the APEX proteome alone, and 27 (28% ) present in the BirA proteome alone. To address the reviewer comments, we compared these percentages to the percentages of total proteins found in both data sets (38%), or the APEX-only (12%) and BirA-only (49%) datasets. As the reviewers suggested, and as is consistent with our finding of more false positives in the BirA proteome (Figure 3), the BirA-only data is underrepresented in the ‘true positives’ list (28%), compared to the proteomic data generally (49%). We have added the following paragraph to the manuscript to describe this analysis:

‘We analysed which of the 97 ‘true positive’ mitochondrial proteins were found in both proteomes, and which were identified in the mtAPEX or mtBirA* proteomes alone (Supplementary file 3). […] This raises the possibility that proteins categorised in the mtBirA* proteome alone contain more false positives than in the shared and mtAPEX-alone categories.’

We have added additional interpretation to our findings in Supplementary file 3 that ‘known’ false positives are enriched for proteins in the mtBirA*-only dataset: “Notably, 14 of these were identified only in the mtBirA* proteome, while 1 was present in both the mtBirA* and mtAPEX proteomes, again consistent with the existence of more false positives in the mtBirA* proteome.”

We have added the following sentence to our analyses on the experimental localisations described in Figure 3.

‘This is consistent with the dataset of proteins identified only in the mtBirA* proteome having more false positives than the other datasets.’

We have also included a comparison of the *T. gondii* mitochondrial proteome with a ‘high confidence’ yeast mitochondrial proteome, which may be consistent with the existence false positives in the mtBirA*-only data set (see response to point 2 below).

Also in this context it is not evident to the reader that the 2 reported mitoproteomes come from two different parasite populations – extracellular (APEX) vs. intracellular (BioID) cultures. Importantly intracellular and extracellular parasites reflect two different biological states with marked changes in mRNA expression when transitioning from one to the other state (Gaji et al., Mol Microbiol. 2011). Parasites show also changes in metabolism when kept extracellular for more than an hour (Lin, S. et al., Int J Parasitol. 2011). Pyruvate carboxylase is known to be transcribed but not translated under some conditions (Nitzsche et al., 2017). In consequence, time scale during which parasites were kept extracellular under APEX labeling/washing/handling conditions could well influence the results and showed be discussed.

Differing mitochondrial proteomes in intra- vs. extra-cellular parasites is an interesting possibility, and we thank the reviewers for pointing this out. It is also conceivable that differences in the proteomes may reflect different protein binding affinities between biotin-phenol and BirA*-catalysed biotinylation. Curiously, 17/20 of the predicted ribosomal proteins or ribosome assembly proteins were identified only in the BirA proteome (Supplementary file 3). One possible explanation is that mitochondrial ribosomes are differentially active in intra- vs extra-cellular parasites. Similarly, we found that 5/6 mitochondrial solute carrier proteins were identified only in the APEX proteome, suggesting either that these are more abundant in extracellular parasites, or that this class of proteins is more accessible to biotin-phenol labelling than to BirA*-mediated biotinylation. Resolving these differences (e.g. by using rapidly labelling BirA* derivatives such as TurboID, in intra- *and* extracellular parasites) may be possible, but are beyond the scope of this study. We have modified the manuscript to note the possibility of different outcomes stemming from differences in the labelling conditions as follows:

‘The mtAPEX labelling was performed over a short timeframe in extracellular parasites, whereas the mtBirA* labelling was performed across 24 hours in intracellular parasites. […] This may reflect either differences in the abundance of these proteins in intra- vs extra-cellular parasites, or in different labelling affinities for these integral membrane proteins between biotin-phenol and BirA*-catalysed biotinylation.’

2) The authors make little use of a database like Mitominer and mentioned analyses in the respective publication (Smith, A. C. and Robinson, A. J. MitoMiner. Nucleic Acids Res 2016). This includes the usage of more than one targeting algorithm like iPSORT, TargetP, MitoFates etc. and compare their outcomes, as it is common use. It will make the statements in the first paragraph of the subsection “Bioinformatic characterisation of the mitochondrial proteome” more convincing.

Following the reviewers suggestions, we analysed our mitochondrial proteome data using MitoFates. We found that MitoFates performed poorly in identifying the predicted mitochondrial proteins stemming from our proteomic analyses in *T. gondii*. Using the same cutoffs as MitoProt (Figure 2—figure supplement 1), we found that 25/421 (6%) of putative mitochondrial proteins had a MitoFates score of >0.9, 63/421 (15%) had a MitoFates score of between 0.5 and 0.9, and the remainder (79%) had a MitoFates score of <0.5. A major factor in the poor performance of MitoFates in predicting *T. gondii* mitochondrial proteins probably results from MitoFates not being developed using *T. gondii* mitochondrial proteins. We do not think these data add much to our manuscript beyond the analysis we have already performed for MitoProt (which, as we already point out in the manuscript, was also not developed using *T. gondii* proteins), and have chosen not to include them in the resubmitted manuscript.

Also, using MiotMiner a more comprehensive comparison to other proteomes should be easy to perform and be informative with regard to a statement like "Our proteome identified most of the proteins 'expected' to localize to the T. gondii mitochondrion suggesting a high level of coverage".

To address this, we have compared our *T. gondii* mitochondrial proteomes to the ‘high-confidence’ yeast mitochondrial proteome described by Morgenstern et al., 2017, which consists of 901 mitochondrial proteins identified in an integrative approach that analysed mitochondrial proteomes, and proteomes of sub-mitochondrial fractions, from cells grown under different growth conditions, using a range of proteomic approaches. We identified 161 homologues to yeast proteins in the *T. gondii* mitochondrial proteome, of which 103 were homologues to proteins localised in the yeast mitochondrion (we include these new data in Supplementary file 1, Tab 5). Of the 58 proteins that had homologues that did not localise to the yeast mitochondrion, 72% were identified in the mtBirA*-only dataset. This may provide further evidence that this dataset is likely to contain more false positives. We have included the following paragraph in the Results section of the manuscript to describe these data:

‘In an additional analysis, we searched for homologs of proteins from the *T. gondii* mitochondrial proteome in a recently published ‘high-confidence’ mitochondrial proteome of yeast (Morgenstern et al., 2017). […] The high proportion of nonmitochondrial homologs suggests that aspects of mitochondrial biology in *T. gondii* may localize elsewhere in other eukaryotes. Notably, however, 42 of the 58 ‘non-mitochondrial’ proteins (72%) were identified in the BirA*-only dataset, which our previous analyses suggest may harbor more false positives. It is conceivable, therefore, that many of the proteins with non-mitochondrial homologs in yeast are false positives.’

3) Subsection “Spatially-restricted biotinylation of mitochondrial matrix proteins”, third paragraph: it is surprising that no mitochondrial labeling in absence of biotin was observed in Figure 1E, given that pyruvate carboxylase is present in mitos (TGME49_284190; with >350 spectra in the author's dataset at ToxoDB) and others observe staining in IFAs with streptavidin (e.g. Jelenska 2001). Please comment. Also, this protein is not part of Supplementary file 3 of known mito proteins, although its mitochondrial localization has been reported (Nitzsche et al., 2017).

The Jelenska study (Figure 1 of that manuscript) as well as other studies that examined the localization of natively biotinylated proteins in *T. gondii* (e.g. Chen et al., 2015) found that the apicoplast-localised acetyl-CoA carboxylases are the major proteins labelled by these approaches. The Jelenska study found that visualization of the mitochondrion (presumably through labelling the mitochondrial pyruvate carboxylase) occurs only upon overexposure during imaging.

We chose not to include pyruvate carboxylase in our analysis of ‘known mitochondrial’ proteins because its natural biotinylation complicates our ability to determine whether it is enriched in experimental vs. control samples (i.e. we ‘expect’ it to be present at high abundance in both experimental and control samples, which, as the reviewers point out, is exactly what we see), thereby biasing our analysis. We have included the following sentence in the description of Supplementary file 3 to make this explicit:

‘Note that natively biotinylated proteins (including the mitochondrially-localized pyruvate carboxylase; (Nitzsche et al., 2017)) were excluded from these analyses because of their biased (i.e. APEX and BirA*-independent) enrichment in control and experimental conditions.’

The latter work also shows PEPCK1 as mitochondrial, whereas here it is indicated as false positive (based on an older publication, Fleige IJP 2008). A more rigorous discussion and evaluation of this data should be included.

We thank the reviewers for pointing this out. We now include PEPCK1 in the ‘true positives’ list of Supplementary file 3.

4) The authors did not detect any of the 3 proteins encoded by the mitochondrial genome, which is a bit curious. Could it be that we still do not have an authentic mtDNA sequence for Toxoplasma, which could affect proteomic detection? The authors should elaborate on the possible reasons why the products of the mitochondrial resident genes remain undetected.

We can think of three possibilities for why we were unable to identify any of the proteins encoded on the mitochondrial genome.

a) All three are highly hydrophobic, and likely deeply embedded in the mitochondrial inner membrane (as part of protein complexes). They may therefore not be able to be biotinylated by the matrix-localised APEX and BirA* proteins. Consistent with this (and as noted later in the results), we were able to identify a possible CoxIII homologue in the ApiCox25 co-immunoprecipitation, a biotin-independent approach (although were unable to detect CoxI in the co-IPs).

b) It is possible that due to the hydrophobic nature of these proteins, they weren’t available for enzymatic digestion, and thus failed to be detected by mass spectrometric analysis. In this regard, it is notable that proteomic analyses of the ATP synthase complex of the parasite were unable to identify the similarly hydrophobic subunit c (Atp9) of the F_o_, 2018).

c) As the reviewers allude, the mitochondrial genome of *T. gondii* has not been published or elsewhere verified, and non-functional copies of the mitochondrial genome have migrated into the genome of *T. gondii*. Proteins with homology to CoxI, CoxIII and cytochrome *b* were included in the proteome list against which we searched our proteomic data, but it is unclear whether these represent the ‘authentic’ mitochondrial proteins, or copies encoded in the nuclear genome.

We have added the following sentences to the manuscript:

‘This is a surprising result. […] It is conceivable, therefore, that the predicted amino acid sequences of these do not match the sequence of the true proteins, which may have prevented their detection in our approaches.’